# Tau, XMAP215/Msps and Eb1 co-operate interdependently to regulate microtubule polymerisation and bundle formation in axons

Ines Hahn[1]*, Andre Voelzmann[1], Jill Parkin[1], Judith B. Fülle[1], Paula G. Slater[2], Laura Anne Lowery[3], Natalia Sanchez-Soriano[4‡]*, Andreas Prokop[1‡]*

**1** The University of Manchester, Manchester Academic Health Science Centre, Faculty of Biology, Medicine and Health, School of Biological Sciences, Manchester, United Kingdom, **2** Department of Biology, Boston College, Chestnut Hill, Massachusetts, United States of America, **3** Department of Medicine, Boston University Medical Center, Boston, Massachusetts, United States of America, **4** Department of Molecular Physiology & Cell Signalling, Institute of Systems, Molecular & Integrative Biology, University of Liverpool, Liverpool, United Kingdom

‡ These authors are joint senior authors on this work.
* Ines.Hahn@manchester.ac.uk (IH); N.Sanchez-Soriano@liverpool.ac.uk (NS-S); Andreas.Prokop@manchester.ac.uk (AP)

**Data Availability Statement:** The authors confirm that all data underlying the findings (underlying numerical data for all graphs and summary statistics) are fully available without restriction. All

## Abstract

The formation and maintenance of microtubules requires their polymerisation, but little is known about how this polymerisation is regulated in cells. Focussing on the essential microtubule bundles in axons of *Drosophila* and *Xenopus* neurons, we show that the plus-end scaffold Eb1, the polymerase XMAP215/Msps and the lattice-binder Tau co-operate interdependently to promote microtubule polymerisation and bundle organisation during axon development and maintenance. Eb1 and XMAP215/Msps promote each other's localisation at polymerising microtubule plus-ends. Tau outcompetes Eb1-binding along microtubule lattices, thus preventing depletion of Eb1 tip pools. The three factors genetically interact and show shared mutant phenotypes: reductions in axon growth, comet sizes, comet numbers and comet velocities, as well as prominent deterioration of parallel microtubule bundles into disorganised curled conformations. This microtubule curling is caused by Eb1 plus-end depletion which impairs spectraplakin-mediated guidance of extending microtubules into parallel bundles. Our demonstration that Eb1, XMAP215/Msps and Tau co-operate during the regulation of microtubule polymerisation and bundle organisation, offers new conceptual explanations for developmental and degenerative axon pathologies.

## Author summary

Axons are the up-to-meter-long processes of nerve cells that form the cables wiring our nervous system. Once established, they must survive for a century in humans. Improper extension of axons leads to neurodevelopmental defects, and age- or disease-related neurodegeneration usually starts in axons. Axonal architecture and function depend on

relevant data are within the paper and its Supporting information files.

**Funding:** This work was made possible through support by the BBSRC to A.P (BB/I002448/1, BB/P020151/1, BB/L000717/1, BB/M007553/1) to N.S.S. (BB/M007456/1, BB/R018960/1), by the Leverhulme Trust to I.H. (ECF-2017-247), by the Deutsche Forschungsgemeinschaft to A.V. (VO 2071/1-1), by NIH to L.A.L (R01 MH109651), and a postdoctoral fellowship from Consejo Nacional de Innovación, Ciencia y Tecnología (https://www.gob.pe/concytec) to P.G.S. The Manchester Bioimaging Facility microscopes used in this study were purchased with grants from the BBSRC (https://bbsrc.ukri.org/), The Wellcome Trust (https://wellcome.org/) and The University of Manchester Strategic Fund (https://www.bmh.manchester.ac.uk/research/support/funding/strategic/). The Fly Facility has been supported by funds from The University of Manchester (https://www.bmh.manchester.ac.uk/research/support/funding/strategic/) and the Wellcome Trust (087742/Z/08/Z; AP). Stocks obtained from the Bloomington Drosophila Stock Center (NIH P40OD018537) were used in this study. The funders had no role in study design, data collection and analysis, decision to publish, or preparation of the manuscript.

**Competing interests:** The authors have declared that no competing interests exist.

bundles of filamentous polymers, called microtubules. These bundles run all along the axonal core, and their disruption correlates with axon decay. How these axonal microtubule bundles are formed and dynamically maintained is little understood. We bridge this knowledge gap by studying how different classes of microtubule-binding proteins may regulate these processes. Here we show how three proteins of very different function, Eb1, XMAP215 and Tau, cooperate intricately to promote the polymerisation processes that form new microtubules during axon development and maintenance. If either protein is dysfunctional, polymerisation is slowed down and newly forming microtubules fail to align into proper bundles. These findings provide new explanations for the decay of microtubule bundles, hence axons. To unravel these mechanisms, we used the fruit fly as a powerful organism for biomedical discoveries. We then showed that the same mechanisms act in frog axons, suggesting they might apply also to humans.

## Introduction

Axons are the enormously long cable-like cellular processes of neurons that wire nervous systems. In humans, axons of $\leq$15μm diameter can be up to two meters long [1,2]. They are constantly exposed to mechanical challenges, yet have to survive for up to a century; we lose ~40% of axons towards high age and far more in neurodegenerative diseases [3–5].

Their growth and maintenance absolutely require parallel bundles of microtubules (MTs) that run all along axons, providing the highways for life-sustaining transport and driving morphogenetic processes. Consequently, bundle decay through MT loss or disorganisation is a common feature in axon pathologies (summarised in [2,6]). Key roles must be played by MT polymerisation, which is not only essential for the *de novo* formation of MT bundles occurring during axon growth in development, plasticity or regeneration, but also to repair damaged and replace senescent MTs during long-term maintenance [7–9]. However, the molecular mechanisms regulating MT polymerisation in axons are surprisingly little understood.

MT polymerisation is primarily understood *in vitro*, where MTs can undergo polymerisation in the presence of nucleation seeds and tubulin heterodimers; the addition of catalytic factors such as CLASPs, stathmins, tau, Eb proteins or XMAP215 can enhance and refine these events [10–16]. However, we do not know whether mechanisms observed in reconstitution assays are biologically relevant in the context of axons [7], especially when considering that none of the above-mentioned factors has genetic links to human neurological disorders on OMIM (Online Mendelian Inheritance in Man), except Tau/MAPT which features primarily with dominant mutations relating to functions less likely to represent its intrinsic MT-regulatory roles [2,17,18].

To identify relevant factors regulating axonal MT polymerisation, we use *Drosophila* primary neurons as one consistent model, which is amenable to combinatorial genetics as a powerful strategy to decipher complex regulatory networks [19]. Our previous loss-of-function studies of 9 MT plus-end-associating factors in these *Drosophila* neurons (CLASP, CLIP190, dynein heavy chain, APC, p150[Glued], Eb1, Short stop/Shot, doublecortin, Lis1) have taken axon length as a crude proxy readout for net polymerisation, mostly revealing relatively mild axon length phenotypes, with the exception of Eb1 and Shot which cause severe axon shortening [20–22].

Here we have incorporated more candidate factors and additional readouts to take these analyses to the next level. We show that three factors, Eb1, XMAP215/Msps and Tau, share a unique combination of mutant phenotypes in culture and *in vivo*, including reduced axonal

MT polymerisation in frog and fly neurons. Our data reveal that the three factors co-operate. Eb1 and XMAP215/Msps act interdependently at MT plus-ends, whereas Tau acts through a novel mechanism: it outcompetes Eb1-binding along MT lattices, thus preventing the depletion of Eb1 pools at polymerising MT plus-ends. By upholding these Eb1 pools, the functional trio also promotes the bundle conformation of axonal MTs through a guidance mechanism mediated by the spectraplakin Shot. Our work uniquely integrates molecular mechanisms into understanding of MT regulation that is biologically relevant for axon growth, maintenance and disease.

## Methods

### Fly stocks

Loss-of-function mutant stocks used in this study were the deficiencies *Df(3R)Antp17* (*tub^def*; removing both *αtub84B* and *αtub84D*; [23,24]), *Df(2L)Exel6015* (*stai^Df*; [25], *Df(3L)BSC553* (*clasp^Df*; Bloomington stock #25116; [20]), *Df(3R)tauMR22* (*tau^Df*; [26]) and the loss-of-function mutant alleles *α-tub84B^KO* (an engineered null-allele; [24]), *chromosome bows^2* (*clasp^2*, an amorph allele; [27]), *Eb1^04524* and *Eb1^5* (two strong loss-of-function mutant alleles; [28]), *futsch^P158* (*MAP1B^-*; a deficiency uncovering the *futsch* locus; [29]), *msps^A* (a small deletion causing a premature stop after 370 amino acids; gift from H. Ohkura), *msps^146* [30], *sentin^ΔB* *short spindle2^ΔB*, (*ssp2^ΔB*; [31]), *tacc^1* (*dTACC^1*; [32]), *shot^3* (the strongest available allele of *short stop*; [21,33]), *stai^KO* [34], *tau^KO* (a null allele; [35]. Gal4 driver lines used were *elav-Gal4* (1^st and 3^rd chromosomal, both expressing pan-neuronally at all stages; [36]), *GMR31F10-Gal4* (Bloomington #49685; expressing in T1 medulla neurons; [37]). Lines for targeted gene expression were *UAS-Eb1-GFP* and *UAS-shot-Ctail-GFP* [22], *UAS-shot^ΔABD-GFP* [38], *UAS-shot^3MTLS\*-GFP* [22], *UAS-dtau-GFP* [26], *UAS-GFP-α-tubulin84B* [39] and further lines generated here (see below).

### *Drosophila* primary cell culture preparation

*Drosophila* primary neuron cultures were done as described previously [40,41]. Stage 11 embryos were treated for 90 s with bleach to remove the chorion, sterilized for ~30 s in 70% ethanol, washed in sterile Schneider's medium containing 20% fetal calf serum (Schneider's/FCS; Gibco), and eventually homogenized with micro-pestles in 1.5 ml centrifuge tubes containing 21 embryos per 100 μl dispersion medium [40] and left to incubate for 4 min at 37°C. Dispersion was stopped with 200 μl Schneider's/FCS, cells were spun down for 4 mins at 650 g, supernatant was removed and cells re-suspended in 90 μl of Schneider's/FCS; 30 μl drops were placed in culture chambers and covered with cover slips. Cells were allowed to adhere to cover slips for 90–120 min either directly on glass or on cover slips coated with a 5 μg/ml solution of concanavalin A, and then grown as a hanging drop culture at 26°C as indicated.

To eliminate a potential maternal rescue of mutants (i.e. reduction of the mutant phenotype due to normal gene product deposition from the wild-type gene copy of the heterozygous mothers in oocytes [42], we used a pre-culture strategy [40,43] where cells were incubated in a tube for 5–7 days before they were plated on coverslips.

For larval cultures, brains from third instar larvae were dissected in PBS (2–3 per cover slip), transferred into Schneider's/FCS medium, washed three times with medium and then processed via homogenisation and dispersion as explained above.

Transfection of *Drosophila* primary neurons was executed as described previously [37]. In brief, 70–75 embryos per 100 μl dispersion medium were used. After the washing step and centrifugation, cells were re-suspended in 100 μl transfection medium [final media containing 0.1–0.5 μg DNA and 2 μl Lipofecatmine 2000 (L2000, Invitrogen)], incubated following

manufacturer's protocols (Thermo Fisher, Invitrogen) and kept for 24 hrs at 26˚C. Cells were then treated again with dispersion medium, re-suspended in culture medium and plated out as described above.

### *Xenopus* primary neuron culture preparation

All experiments were approved by the Boston College Institutional Animal Care and Use Committee and performed according to national regulatory standards. *Xenopus* primary neuron cultures were obtained from embryonic neural tubes. Eggs collected from female *X. laevis* frogs were fertilised *in vitro*, dejellied and cultured following standard methods [44]. Embryos were grown to stage 22–24 [45], and neural tubes were dissected as described [46]. Three neural tubes were transferred for 10 minutes to an Eppendorf tube containing 150 μl CMF-MMR (0.1 M NaCl, 2.0 mM KCl, 1.0 mM EDTA, 5.0 mM HEPES, pH 7.4), centrifuged at 1000 g for 5 min, and 150 μl of Steinberg's solution [58 mM NaCl, 0.67 mM KCl, 0.44 mM $Ca(NO_3)_2$, 1.3 mM $MgSO_4$, 4.6 mM Tris, pH 7.8] was added to the supernatant to follow with the tissue dissociation using a fire-polished glass Pasteur pipet. Cells were seeded on 60 mm plates pretreated with 100 μg/ml poly-L-lysine and 10 μg/ml laminin; after 2 hrs the medium was replaced by plating culture medium (50% Ringer's, 49% L-15 medium, 1% fetal bovine serum, 25 ng/μl NT3 and BDNF, plus 50 μg/ml penicillin/streptomycin and gentamycin, pH 7.4 and filter-sterilized) and kept for 24 hr before imaging.

The embryos were injected four times in dorsal blastomeres at two-to-four cell stage with 6 ng of the validated XMAP215 morpholino (MO; [47]), 10 ng of the validated tau MO [48], and/or 5 ng of a newly designed splice site MO for EB3 (3´CTCCCAATTGTCACCTACTTT GTCG5´; for verification see S5 Fig), in order to obtain a 50% knockdown of each.

### Dissection of adult fly heads

For *in vivo* studies, brain dissections were performed in Dulbecco's PBS (Sigma, RNBF2227) after briefly sedating them on ice. Dissected brains with their laminas and eyes attached were placed into a drop of Dulbecco's PBS on MatTek glass bottom dishes (P35G1.5-14C), covered by coverslips and immediately imaged.

### Immunohistochemistry

Primary fly or frog neurons were fixed in 4% paraformaldehyde (PFA; in 0.05 M phosphate buffer, pH 7–7.2) for 30 min at room temperature (RT). For anti-Eb1 and anti-GTP-tubulin staining, cells were fixed for 10 mins at -20˚C in +TIP fix (90% methanol, 3% formaldehyde, 5 mM sodium carbonate, pH 9; stored at -80˚C and added to the cells [49]), then washed in PBT (PBS with 0.3% TritonX). Antibody staining and washes were performed with PBT. Staining reagents: anti-tubulin (clone DM1A, mouse, 1:1000, Sigma; alternatively, clone YL1/2, rat, 1:500, Millipore Bioscience Research Reagents); anti-DmEb1 (gift from H. Ohkura; rabbit, 1:2000; [28]); anti-GTP-tubulin (hMB11; human, 1:200; AdipoGen; [50]); anti-Shot (1:200, guinea pig; [51]); anti-Elav (Elav-7E8A10; rat, 1:1000; Developmental Studies Hybridoma Bank, The University of Iowa, IA, USA; [52]); anti-GFP (ab290, Abcam, 1:500); Cy3-conjugated anti-HRP (goat, 1:100, Jackson ImmunoResearch); F-actin was stained using phalloidin conjugated with TRITC/Alexa647, FITC or Atto647N (1:100 or 1:500; Invitrogen and Sigma). Specimens were embedded in ProLong Gold Antifade Mountant (ThermoFisher Scientific).

## Microscopy and data analysis

Standard imaging was performed with AxioCam 506 monochrome (Carl Zeiss Ltd.) or Matrix-Vision mvBlueFox3-M2 2124G digital cameras mounted on BX50WI or BX51 Olympus compound fluorescent microscopes. For the analysis of *Drosophila* and *Xenopus* primary neurons, we used the following parameters:

Axon length was measured from cell body to growth cone tip using the segmented line tool of ImageJ [22,43].

Degree of disorganised MT curling in axons was measured as "MT disorganisation index" (MDI) described previously [37,41]; in short: the area of disorganised curling was measured with the freehand selection in ImageJ; this value was then divided by axon length (see above) multiplied by 0.5 μm (typical axon diameter, thus approximating the expected area of the axon if it were properly bundled).

Eb1 comet amounts were approximated by using the product of comet mean intensity and length. For this, a line was drawn through each comet (using the segmented line tool in Fiji) to determine length as well as mean staining intensity of Eb1 or GTP-tub in fixed *Drosophila* and MACF43::GFP in movie stills of *Xenopus* neurons.

To measure MT curling in the optic lobe of adult flies, *GMR31F10-Gal4* (Bloomington #49685) was used to express *UAS-α-tubulin84B-GFP* [39] in a subset of lamina axons which project within well-ordered medulla columns [53]. Flies were left to age for 26–27 days (about half their life expectancy) and then their brains were dissected as explained above and immediately imaged with a 3i Marianas Spinning Disk Confocal Microscope at the ITM Biomedical imaging facility at the University of Liverpool. A section of the medulla columns comprising the 4 most proximal axonal terminals was used to quantify the number of swellings and regions with disorganised curled MTs.

To measure MT polymerisation dynamics in *Drosophila* neurons [54], movies were collected on an Andor Dragonfly200 spinning disk upright confocal microscope (with a Leica DM6 FS microscope frame) and using a *100x/1.40 UPlan SAPO (Oil)* objective. Samples where excited using 488 nm (100%) and 561 nm (100%) diode lasers via Leica GFP and RFP filters respectively. Images where collected using a Zyla 4.2 Plus sCMOS camera with a camera gain of 1x. The incubation temperature was set to 26˚C. Time lapse movies were constructed from images taken at 1 s intervals for 1 min. To measure comet velocity and lifetime, a line was drawn that followed the axon using the segmented line tool in Fiji/ImageJ. A kymograph was then constructed from average intensity in Fiji using the KymoResliceWide macro (Cell Biology group, Utrecht University) and events scored via the Velocity Measurement Tool Macro (Volker Baecker, INSERM, Montpellier, RIO Imaging; J. Rietdorf, FMI Basel; A. Seitz, EMBL Heidelberg). For each condition at least 15 cells were analysed in ≥2 independent repeats.

To assess EB1 comet dynamics and comet amounts in *Xenopus* neurons, 300 pg of MACF43-Ctail::GFP (an Eb protein-binding 43-residue fragment derived from the C-terminal regions of hMACF2/human microtubule actin crosslinking factor 2; Fig 3F-3F'''; [55,56]), was co-injected with the MO. Time lapse imaging of *Xenopus* primary cultures was performed with a CSU-X1M 5000 spinning-disk confocal (Yokogawa, Tokyo, Japan) on a Zeiss Axio Observer inverted motorized microscope with a Zeiss 63× Plan Apo 1.4 numerical aperture lens (Zeiss, Thornwood, NY). Images were acquired with an ORCA R2 charge-coupled device camera (Hamamatsu, Hamamatsu, Japan) controlled with Zen software. Time lapse movies were constructed from images taken at 2 s intervals for 1 min. MACF43 comet velocity and lifetime were analysed with plusTipTracker software. The same parameters were used for all movies: maximum gap length, eight frames; minimum track length, three frames; search

radius range, 5–12 pixels; maximum forward angle, 50˚; maximum backward angle, 10˚; maximum shrinkage factor, 0.8; fluctuation radius, 2.5 pixels; and time interval 2 s.

Images were derived from at least 3 independent experimental repeats performed on different days, for each of which at least 2 independent culture wells were analysed by taking a minimum of 20 images per slide. For statistical analyses, Kruskal–Wallis one-way ANOVA with *post hoc* Dunn's test or Mann–Whitney Rank Sum Tests (indicated as $P_{MW}$) were used to compare groups, r and p-value for correlation were determined via non-parametric Spearman correlation analysis (tests showed that data are not distributed normally). All raw data of our analyses are provided (S1–S14 Data).

### Western blot analysis of *Xenopus* embryos

For protein extraction, 10 embryos were transferred to a centrifuge tube with 800 μl lysis buffer (50 mM Tris pH 7.5, 5% glycerol, 0.2% IGEPAL/NP-40, 1 mM EDTA, 1.5 mM $MgCl_2$, 125 mM NaCl, 25 mM NaF, 1 mM $Na_3VO_4$), homogenised with a sterile pestle and, after 10 mins, centrifuged at 13,000 rpm for 15–20 min. The supernatant was collected and the protein concentration determined with the Micro BCA Protein Assay Kit (Thermo Fisher Scientific). 80 μg protein were loaded into a 10% SDS gel and stained with anti-Tau (clone Tau46, T9450, mouse, 1:1000, Sigma-Aldrich).

### Molecular biology

To generate the *UAS-msps$^{FL}$-GFP* (aa1-2050) and *UAS-msps$^{ΔCterm}$* (aa1-1322) constructs, eGFP was PCR-amplified from pcDNA3-EGFP and *msps* sequences from cDNA clone *LP04448* (DGRC; FBcl0189229) using the following primers:

*msps$^{FL}$* and *msps$^{ΔCterm}$* fw: *GAATAGGGAATTGGGAATTCGTTAGGCGCGCCAACATGGCC GAGGACACAGAGTAC*

*msps$^{FL}$* rev: *CAAGAAAGAGAATCATGCCCAAGGGCCCGGTAGCGGCAGCGGTAGCGTGA GCAAGGGCGAG*

*msps$^{ΔCterm}$* rev: *GATGGAGGGTCTAAAATCGCATATGGGTAGCGGCAGCGGTAGCGTGAG CAAGGGCGAGGAG*

*eGFP* fw: *GAGAATCATGCCCAAGGGCCCGGTAGCGGCAGCGGTAGCGTGAGCAAGGGC GAGGAGCTG*

*eGFP* rev: *CTCTCGGCATGGACGAGCTGTACAAGTAGGCGGCCGCCTCGAGGGTACCTCTAGAG*

The *msps* and *eGFP* sequences were introduced into *pUAST-attB* via Gibson assembly (ThermoFisher) using *EcoRI* and *XhoI*. To generate transgenic fly lines, *pUAST-attB* constructs were integrated into *PBac{yellow[+]-attP-9A}VK00024* (Bloomington line #9742) via PhiC31-mediated recombination (outsourced to Bestgene Inc).

## Results

### Eb1, Msps/XMAP215 and Tau share the same combination of loss-of-function phenotypes in axons

To identify factors relevant for axonal MT polymerisation, we performed a detailed loss-of-function study with a selection of candidate factors suggested by the literature [7]. We included the MT plus-end-associating factors Eb1, Shot, CLASP/Chb (Chromosome bows) and the

XMAP215 homologue Msps (Mini spindles). As MT shaft-binding candidates, we chose Tau and the Map1b homologue Futsch. To explore the impact of tubulin availability on polymerisation, we used α1-tubulin (αtub84B, the predominant α-tubulin expressed in the fly nervous system; FlyAtlas 2, University of Glasgow, UK) and Stathmin (a promoter of tubulin pools; [25,57]).

We analysed primary neurons derived from animals carrying loss-of-function mutations of these genes. To exclude that phenotypes are masked by maternal contribution (which is wild-type gene product deposited in the eggs by the heterozygous mothers), we used two strategies [40,43]: First, where possible, we analysed larval neurons, i.e. primary neurons derived from late larval brains and cultured for 18 HIV (hours *in vitro*). Second, if mutants did not reach larval stages, we used pre-cultured neurons, i.e. embryo-derived neurons that were kept for 5–7 days in pre-culture to deplete maternal product and then plated and grown for 12 HIV. In all cases, primary neurons were immuno-stained either for endogenous tubulin to assess axon length, or for endogenous Eb1 protein to gain a first insight into the polymerisation state of axonal MTs.

Except for Shot and Futsch, loss of all other factors displayed a significant reduction in the number of Eb1-positive plus-end comets (Fig 1A–1D and 1I and S1A Fig). In addition, we measured the mean intensities and mean lengths of Eb1 comets and used multiplication of these two parameters to approximate Eb1 amounts at MT plus-ends. The strong hypomorphic $Eb1^{04524}$ mutant allele is known to display severe, but not complete reduction of protein levels [28]; accordingly, Eb1 amounts at MT plus-ends were severely, but not completely diminished in neurons mutant for this allele (Fig 1D and 1J and S1B Fig). Out of the remaining seven candidate factors, only two further genes showed the same qualitative mutant phenotype: $msps^{A/A}$ showed a reduction almost as strong as $Eb1^{04524/04524}$, and $tau^{KO/KO}$ mutant neurons showed a milder but reliable Eb1 depletion (Fig 1B, 1C and 1J). In all cases, the drop in Eb1 amounts was to almost equal parts due to reductions in comet length and intensity (S2A and S2B Fig), and correlated well with reduced comet velocities and lifetimes when assessed in live imaging experiments (Fig 1K and 1L and S2C–S2E Fig; using Shot-Ctail::GFP as readout for plus-end dynamics; [22]). Our data demonstrate therefore that comparable comet length/velocity correlations made *in vitro* [58] are relevant in the cellular context of axons.

To assess whether reduced comet velocities or numbers correlate with impaired axon growth, we performed tubulin staining and measured axon length. Out of the eight factors, all but Futsch showed a decrease in axon length ranging between 10 and 43% when compared to parallel control cultures with wild-type neurons (Fig 1E–1H and 1M and S1C Fig). In these experiments, Tau-, Msps-, Shot- or Eb1-deficient axons showed areas of prominent disintegration of axonal MT bundles where MTs displayed intertwined, criss-crossing curls which can be quantified via the MT disorganisation index (MDI; see Microscopy And Data Analysis; white arrowheads in Fig 1F–1H and 1N and S1D Fig). This MT curling phenotype was known for Shot and Eb1 [22], but unexpected for Tau and Msps.

Taken together, out of eight candidate factors, loss of the MAP1B homologue Futsch was the only condition showing no obvious defects. In contrast, loss of Eb1, the XMAP215 homologue Msps and Tau stood out by showing the same phenotypic pattern: reductions in comet numbers, comet velocities, Eb1 amounts and axon lengths, as well as a MT curling phenotype shared with loss of Shot. These five phenotypes appeared consistently milder in *tau* than *Eb1* or *msps* mutant neurons. Our observations raised the question as to why these three factors display such a striking pattern of collective phenotypes.

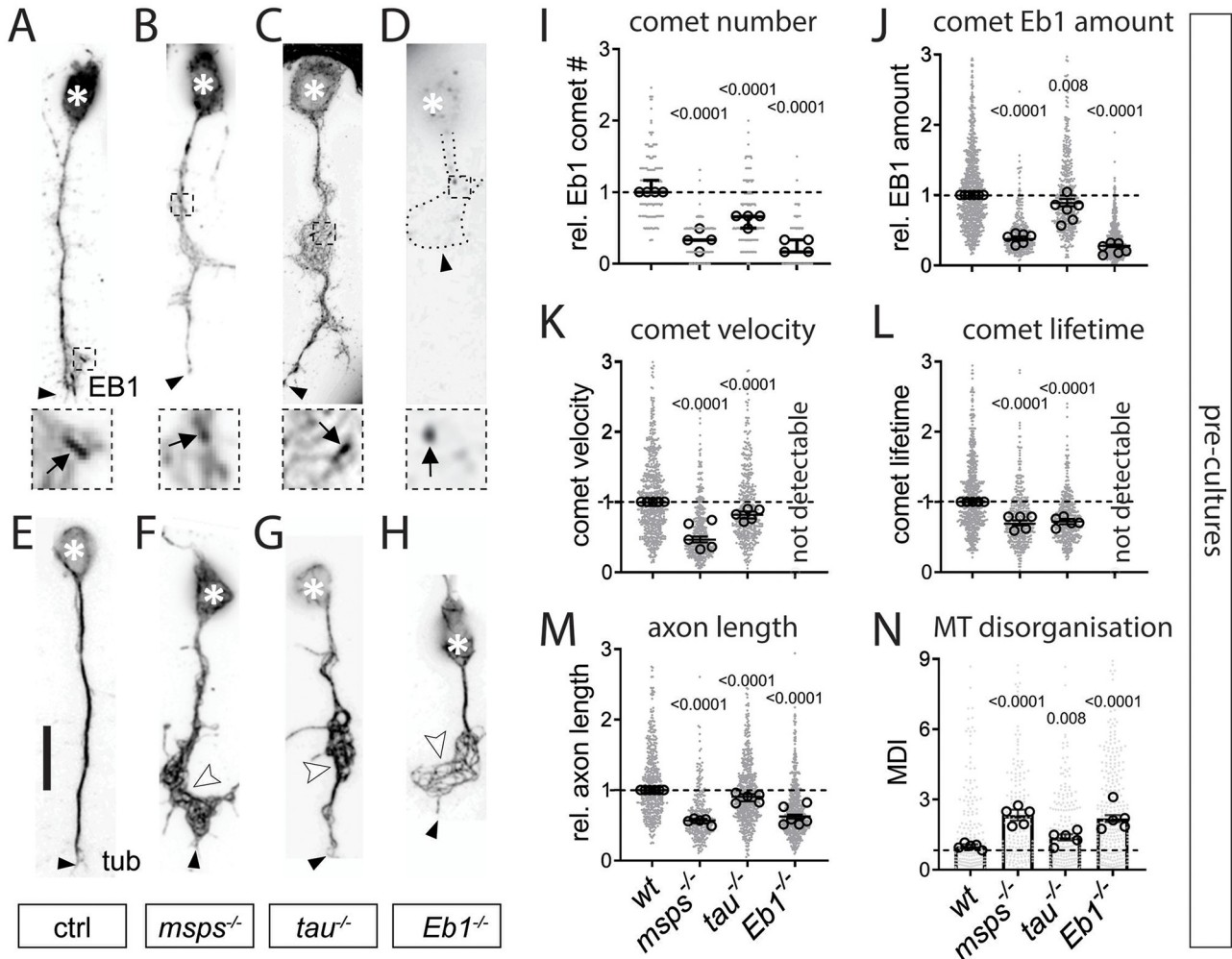

**Fig 1. Eb1, Msps and Tau share the same combination of axonal loss-of-function phenotypes in *Drosophila* primary neurons. A-H)** Images of representative examples of embryonic primary neurons pre-cultured up to 6 days (to deplete maternal gene product; see *Drosophila* Primary Cell Culture Preparation) and either immuno-stained for Eb1 (top) or for tubulin (bottom); neurons were either wild-type controls (ctrl) or carried the mutant alleles *msps^1^*, *tau^KO^* or *Eb1^04524^* in homozygosis (from left to right); asterisks indicate cell bodies, black arrow heads the axon tips, white arrow heads point at areas of MT curling, dashed squares in A-D are shown as 3.5-fold magnified close-ups below each image with black arrows pointing at Eb1 comets; the axonal outline in D is indicated by a dotted line; scale bar in A represents 15 μm in all images. **I-N)** Quantification of different parameters (as indicated above each graph) obtained from pre-cultured embryonic primary neurons with the same genotypes as shown in A-H. Data were normalised to parallel controls (dashed horizontal line) and are shown as median ± 95% confidence interval (I-M) or mean ± SEM (N); data points in each plot, taken from at least two experimental repeats consisting of 3 replicates each; large open circles in graphs indicate median/mean of independent biological repeats. P-values obtained with Kruskall-Wallis ANOVA test for the different genotypes are indicated in each graph. For raw data see S1 Data.

## Eb1, Msps and Tau genetically interact during axonal MT regulation both in culture and *in vivo*

To investigate potential functional relationships between Eb1, Msps and Tau, we performed genetic interaction studies. We used heterozygous conditions (i.e. one mutant and one normal copy) of genes to reduce their respective protein levels. If reduced protein levels of different genes combined in the same neurons cause a phenotype, this suggests that they are likely to function in a common pathway. For our studies, we used larval neurons and five different

readouts: axon length (Fig 2A), MT curling (Fig 2A'), Eb1 amounts at MT plus-ends (Fig 2A"), comet numbers (S3A Fig) and comet dynamics (S3B Fig).

To assess the baseline, we analysed single-heterozygous mutant neurons ($Eb1^{04524/+}$, $msps^{A/+}$ or $tau^{KO/+}$), none of which displayed any phenotypes. Certain double-combinations of these heterozygous conditions in the same neurons ($Eb1^{04524/+}$ $msps^{A/+}$ and $tau^{KO/+}$ $msps^{A/+}$) displayed a mild but significant reduction in Eb1 amounts at MT plus-ends and a trend towards stronger MT curling. Strong enhancement of the phenotypes for all five read-outs was observed in $Eb1^{04524/+}$ $msps^{A/+}$ $tau^{KO/+}$ triple-heterozygous neurons, suggesting that the three factors are functionally linked when regulating MT plus-ends, axon growth and MT organisation (Fig 2A' and 2A" and S3A and S3B Fig).

The triple-heterozygous condition had a similar effect in fly brains *in vivo* when using axons of T1 medulla neurons in the adult optic lobe as readouts. In control flies, analysed 26–27 days after eclosure, axons of T1 neurons contained prominent MT bundles labelled by α-tubulin::GFP (Fig 3A; [37]). In contrast, T1 axons of triple-heterozygous mutant flies aged in parallel, showed a strong increase in areas where MTs become unbundled and twisted and axons displayed prominent swellings often containing curled MTs (Fig 3A–3D). These data strongly suggest that the co-operating network of Eb1, Msps and Tau is relevant *in vivo*.

## Eb1, Msps and Tau interaction is evolutionarily conserved in *Xenopus* neurons

To assess potential evolutionary conservation, we evaluated whether Eb1, Msps and Tau interact also in frog primary neurons. In the frog *Xenopus laevis*, *mapt* (*microtubule associated protein tau*) is the only *tau* homologue, *XMAP215* (*ckap5/cytoskeleton associated protein 5*) the only *msps* homologue, and *EB3* (*mapre3/microtubule associated protein RP/EB family member 3*) is the only one of three *Eb1* homologues that is prominently expressed in the nervous system [59,60]. We used morpholinos against these three genes. Similar to our strategy in *Drosophila*, we analysed MTs by staining for endogenous tubulin (Fig 3E–3E""), and measured Eb3 comet amounts in live movies using the Eb protein-binding peptide MACF43-Ctail::GFP as readout (comparable to Shot-Ctail::GFP used in Fig 1K and 1L; see Microscopy And Data Analysis; [55,56]).

To approximate heterozygous mutant conditions used in our *Drosophila* experiments, we adjusted morpholino concentrations to levels that achieved knock-down of each of the three genes to ~50% (S5A and S5B Fig and [47,56]). Individual knock-downs to approximately 50% did not cause prominent decreases in MACF43::GFP comet amounts or increases in MT curling; but when knock-down of all three factors was combined in the same neurons, we found a reduction in MACF43::GFP comet amounts to 60% and a 4.8 fold increase in MT bundle disintegration and curling (Fig 3E"', 3F"", 3G and 3H and S5C and S5D Fig).

Together, the three genes appear to functionally interact also in *Xenopus*, suggesting evolutionary conservation of the underpinning mechanisms.

## Eb1, Msps and Tau form an interdependent functional network with key roles played by Eb1

To investigate the underlying mechanisms, we next asked whether the functions of the three factors are hierarchically and/or interdependently organised, or whether they regulate the assessed MT properties through independent parallel mechanisms. To distinguish between these possibilities, we performed epistasis experiments [61]. For this, we combined homozygous mutant conditions of the three mutant alleles in the same neurons ($Eb1^{04524/04524}$ $msps^{A/A}$ $tau^{KO/KO}$) and asked whether this condition enhances phenotypes over single-homozygous

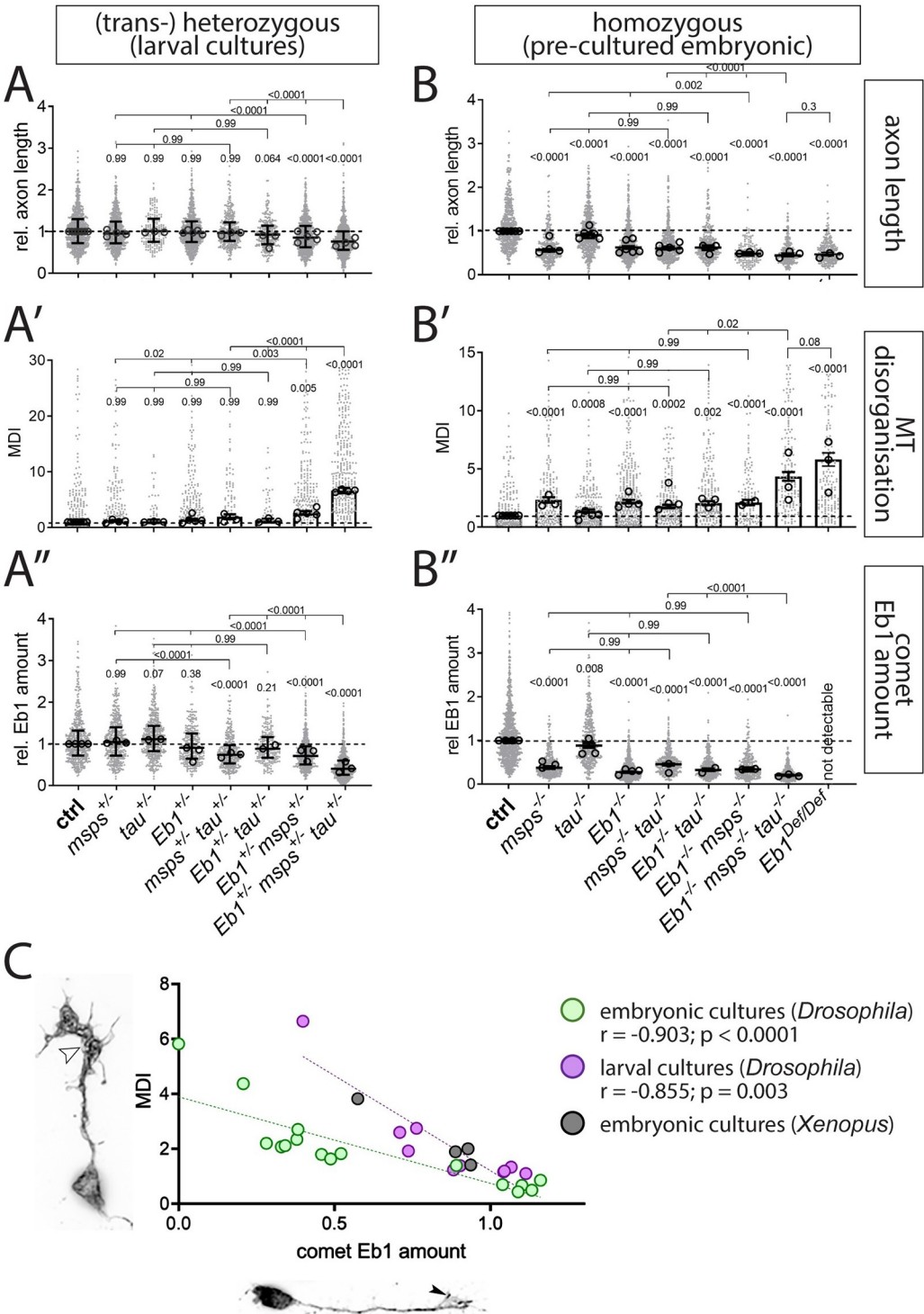

**Fig 2. *Eb1*, *tau* and *msps* interact genetically. A-B")** Axon length, MT curling and Eb1 amount (as indicated on the right), for primary neurons displaying heterozygous (A-A", larval cultures) and homozygous (B-B', embryonic 6d pre-cultures') mutant conditions, alone or in combination. Data were normalised to parallel controls (dashed horizontal lines) and are shown as scatter dot plots with median ± 95% confidence interval (A, A",B, B") or bar chart with mean ± SEM (A', B') of at least two independent repeats with 3 replicates each; large open circles in graphs indicate median/mean of independent biological repeats. P-values above data points/bars were obtained with Kruskall-Wallis ANOVA tests; used alleles: *msps^A^*, *tau^KO^*, *Eb1^04524^*. **C)** The graph compares Eb1 amounts at MT plus-ends with the degree of MT curling (MT disorganisation index/MDI) for a range of genetic conditions used in this work: green dots show data from pre-cultured embryonic neurons (B' vs. B"), purple dots show comparable data obtained from larval primary neurons (A' vs. A"); in

addition, green/purple dots contain data from Fig 1J and 1N and S1B and S1D Fig; black dots show similar data obtained from primary *Xenopus* neurons (Fig 3G and 3H); r and p-value determined via non-parametric Spearman correlation analysis; see further detail of these correlations in S4 Fig. For raw data see S2 Data.

conditions (suggesting parallel mechanisms), or whether they show no further increases (suggesting a hierarchical organisation).

Since the homozygous mutant conditions do not all survive into the late larval stage, we used embryonic pre-cultured neurons for these experiments. As reference we used

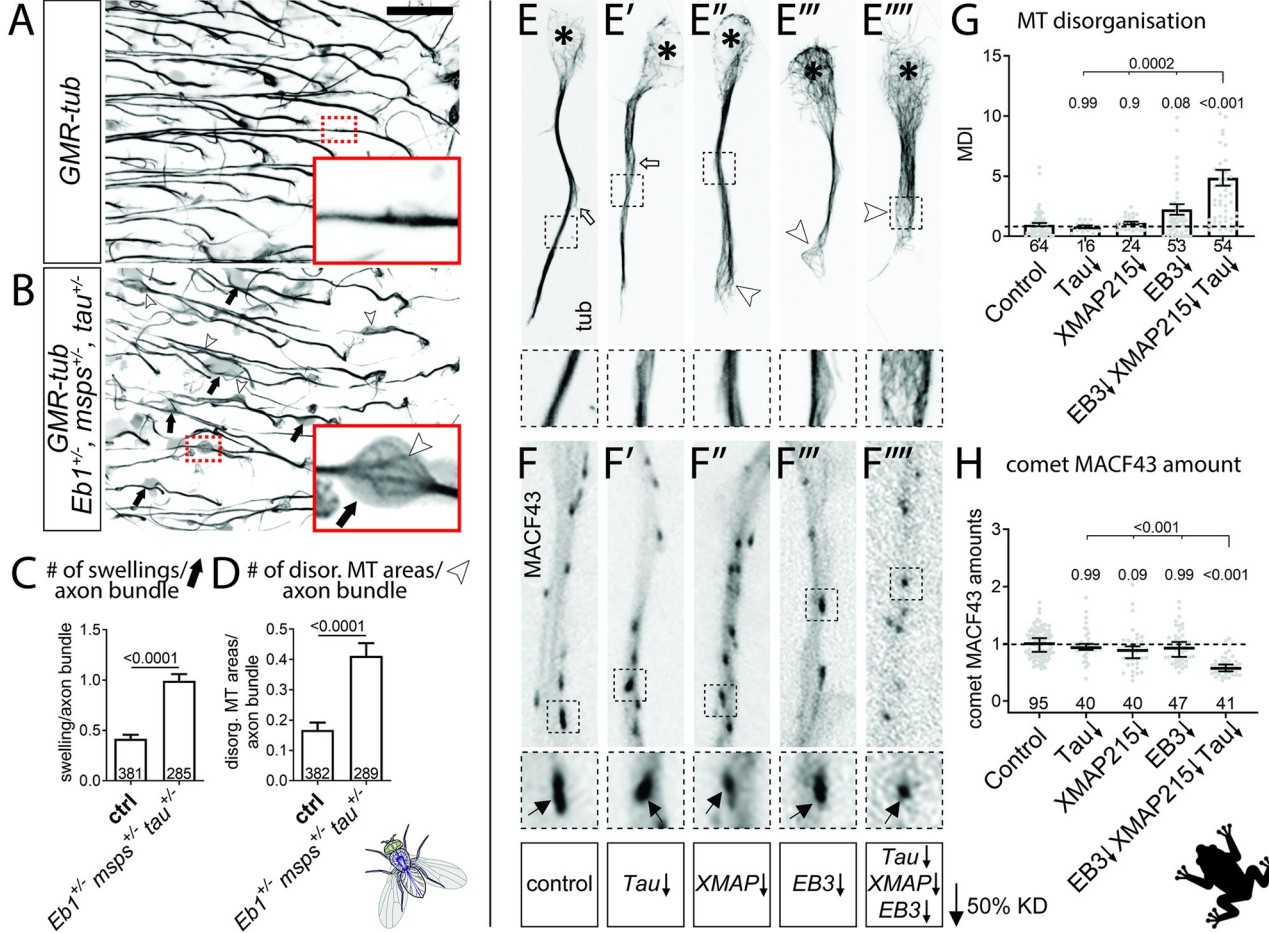

**Fig 3. Eb1, Msps and Tau functionally interact in the fly brain and in frog primary neurons. A,B**) Medulla region of adult brains at 26–27 days after eclosure, all carrying the *GMR31F10-Gal4* driver and *UAS-GFP-α-tubulin84B* (*GMR-tub*) which together stain MTs in a subset of lamina neuron axons that terminate in the medulla; the further genetic background is either wild-type (A) or triple-heterozygous (*Eb1*[04524/+] *msps*[A/+] *tau*[KO/KO]; B); white/ black arrows indicate axonal swellings without/with MT curling; rectangles outlined by red dashed lines are shown as 2.5 fold magnified insets where white arrow heads point at disorganised curling MTs. **C,D**) Quantitative analyses of specimens shown in A and B: relative number of total swellings per axon (C) and of swellings with MT curling per axon (D); bars show mean ± SEM; P values from Kruskal–Wallis one-way tests are given above each column, merged sample numbers (i.e. individual axon bundles) from at least two experimental repeats at the bottom of each bar. **E-E'''**) Primary *Xenopus* neurons stained for tubulin (tub): asterisks indicate cell bodies, white arrows indicate unbundled MTs, white arrowheads unbundled areas with MT curling. **F-F'''**) *Xenopus* neurons labelled with MACF43::GFP (MACF43): black arrows point at comets (visible as black dots). In E-F''', black-stippled squares in overview images are shown as 2.5 fold magnified close-ups below; ↓ behind gene symbols indicates 50% knock-down thus approximating heterozygous conditions. **G,H**) Quantification of specimens shown in E-F'''with respect to MT curling (G) and comet amount of MACF43::GFP (H); data were normalised to parallel controls (dashed horizontal lines) and are shown as mean ± SEM (G) or median ± 95% confidence interval (H); merged sample numbers from at least two experimental repeats are shown at the bottom, P-values obtained with Kruskall-Wallis ANOVA tests above data points/bars. The scale bar in A represents 15 μm in A,B and 20 μm in E-F'''. For raw data see S3 Data.

$Eb1^{04524/04524}$, $msps^{A/A}$ and $tau^{KO/KO}$ single mutant neurons; since $Eb1^{04524}$ is a strong but not a total loss-of-function allele [28], we added neurons homozygous for $Df(2R)Exel6050$ (from now referred to as $Eb1^{Df}$), a deficiency uncovering the entire $Eb1$ locus (see Fly Stocks). The order of phenotypic strengths in these pre-cultured homozygous mutant neurons consistently was $Eb1^{Df/Df} > Eb1^{04524/04524} \geq msps^{1/1} > tau^{KO/KO}$ for all parameters assessed (Fig 2; see Discussion). The strong phenotypes of the $Eb1^{Df/Df}$ allele provided therefore a new reference bar, displaying values that were not excelled or even reached by any combination of homozygous mutants, including the triple-homozygous mutant neurons (Fig 2B–2B").

These results are consistent with a model where the three proteins co-operate interdependently, through a modality in which Tau has the weakest contribution and Eb1 is the most essential factor. This is also suggested when extracting data from a range of different genetic conditions analysed throughout this study and plotting their Eb1 amounts at MT plus-ends against the degree of MT curling in each condition (S4A Fig). This plot revealed a highly significant inverse correlation between Eb1 comet amounts and the degree of MT curling (Fig 2C and S4B Fig), suggesting that Eb1 is the key factor within the functional-trio linking out to MT bundle promotion.

## Eb1 and Msps depend on each other for MT plus-end localisation

We next investigated the mechanistic links between the three proteins, starting with Msps and Eb1. When co-expressing Msps::GFP together with Eb1::RFP in *Drosophila* embryonic primary neurons, both proteins prominently localised at the same MT plus-ends, with Msps::GFP localising slightly distal to Eb1 (Fig 4A–4A" and S1 Movie). Their localisation at the same MT plus-ends was even clearer in kymographs of live movies where both proteins remained closely associated during comet dynamics (Fig 4B and 4B'). These findings agree with localisation data for Eb proteins and XMAP215 *in vitro* [62], suggesting that the same mechanisms apply in the cellular context of axons and might explain the linked phenotypes we observed for the two factors. This was further supported by their co-dependence for achieving prominent plus-end localisation:

We already showed that loss of Msps causes severe Eb1 depletion and comet shortening at MT plus-ends (Fig 1B and 1D and S1B Fig). This is likely due to the role of Msps as a tubulin polymerase [11] which helps to sustain a prominent GTP-tubulin cap to which EB1 can bind [14]. To test this, we stained $msps^{1/1}$ mutant neurons for GTP-tubulin, and found that GTP caps displayed reduced amounts comparable to the shortened Eb1 comets (Fig 1J *vs*. S8C Fig). This result, together with our finding of reduced comet velocity in $msps^{1/1}$ mutant neurons (Fig 1K), is consistent with the hypothesis that Msps-dependent polymerisation increases Eb1 amounts at MT plus-ends (see Discussion).

*Vice versa*, we studied the localisation of Msps::GFP in $Eb1^{04524/04524}$ mutant neurons and found a severe depletion of Msps at growing MT plus-ends when compared to controls (Fig 4C and 4C' and S2 and S3 Movies). These results suggest that Msps/XMAP215 can bind MT plus-ends independently, but that its binding is enhanced if Eb1 is present (see also [14,62]). One mechanism through which Eb1 might recruit Msps/XMAP215 at MT plus-ends is through adaptors such as SLAIN in vertebrates, or TACC (Transforming acidic coiled-coil protein) or Sentin in non-neuronal *Drosophila* cells [11,15,47,63–66]. However, our functional studies using homozygous mutant neurons carrying $tacc^1$ and $sentin^{\Delta B}$ loss-of-function mutant alleles failed to display axon shortening or MT curling, thus arguing against a prominent role of these factors during Eb1-dependent Msps recruitment (details in S6A and S6B Fig). To further substantiate these findings, we generated a $msps^{\Delta Cterm}$-GFP construct which lacks the C-terminal domain essential for the interaction with adaptors (Fig 4D; [67,68]).

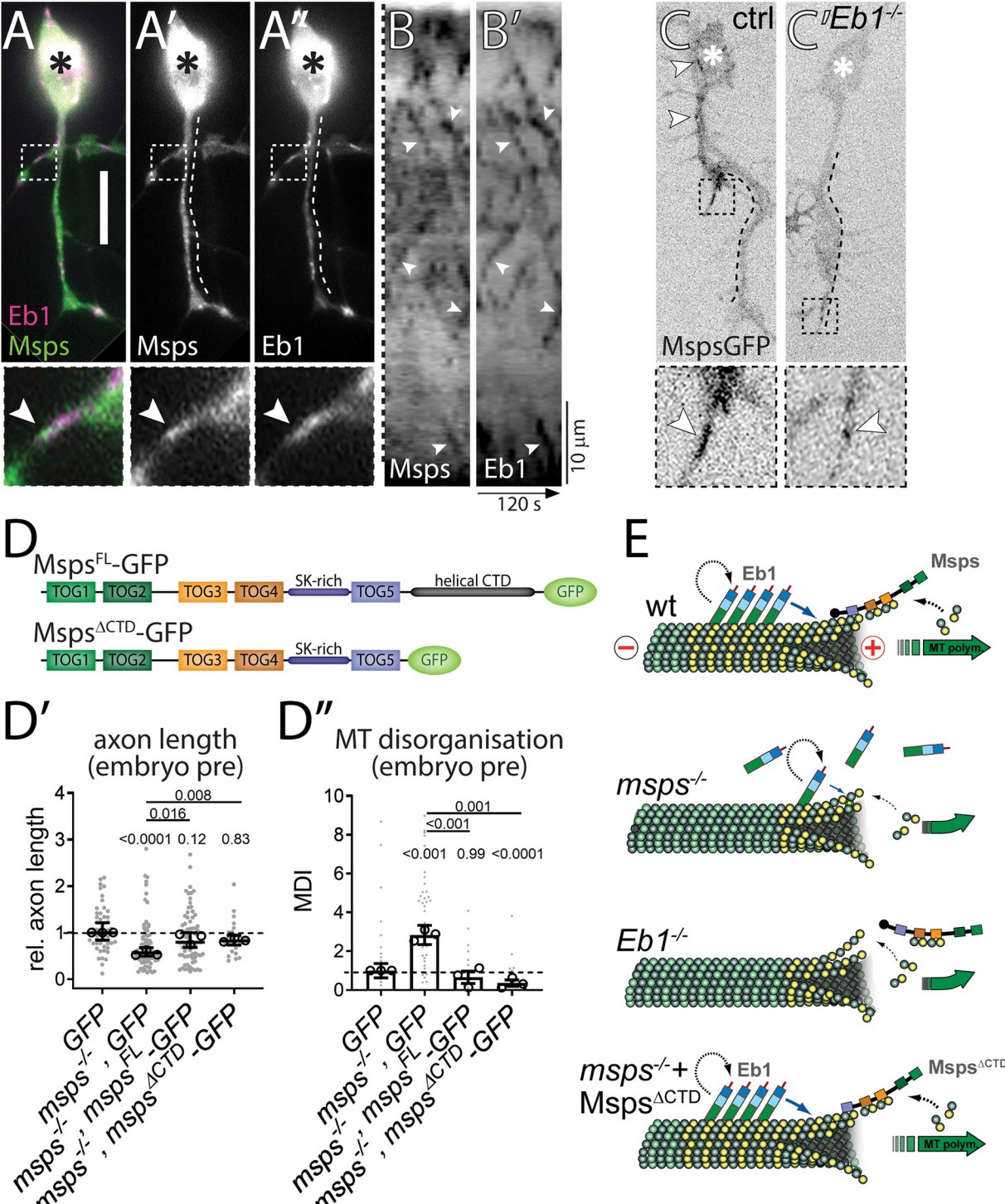

**Fig 4. Eb1 and Msps depend on each other for MT plus-end localisation. A-A")** Primary neurons at 6HIV co-expressing Eb1::mCherry (magenta, Eb1) and Msps^FL^::GFP (green, Msps) and imaged live; asterisks indicate somata, scale bar represents 10µm, dashed boxes indicate the positions of the 3.5-fold magnified close-ups shown at the bottom with arrowheads pointing at the position of Msps::GFP accumulation (same in C,C'). **B,B')** Kymograph of live movies (as in A-A") with the dashed line on the left representing a straightened version of the dashed lines shown in A' and A" (i.e. the length of the axon; proximal at the top) and the x-axis indicating time; arrowheads point at trajectories of Msps and Eb1 which are almost identical. **C)** Primary neurons expressing Msps::GFP and imaged live, either displaying wild-type background (ctrl) or being homozygous mutant for *Eb1^04524^* (*Eb1^-/-^*); white arrowheads point at Msps::GFP comets which are much smaller in the mutant neurons. **D)** Schematic representations of Msps^FL^::GFP and Msps^ΔCTD^::GFP. **D',D")** Graphs displaying axon length and MT curling (as indicated) for pre-cultured embryonic primary neurons expressing GFP or Msps::GFP constructs via the *elav-Gal4* driver, either in wild-type or *msps^A/1^*

mutant background; data were normalised to parallel controls (dashed horizontal lines) and are shown as median ± 95% confidence interval (D')
or mean ± SEM (D") from at least two experimental repeats; large open circles in graphs indicate median/mean of independent biological
repeats. P-values obtained with Kruskall-Wallis ANOVA tests are shown above data points/bars. **E)** Model view of the results shown here and in
Fig 1; note that yellow circles represent GTP-tubulin which mediates the binding of Eb1; for further explanations see main text and Discussion.
For raw data see S4 Data.

When $msps^{\Delta Cterm}$-GFP or $msps^{FL}$-GFP full length controls were transfected into $msps^{A/146}$
mutant neurons, we found that both protein variants were similarly able to improve axon
length and MT curling defects (Fig 4D' and 4D"), thus likewise arguing against the require-
ment of adaptors to recruit Msps to MT plus-ends in fly neurons.

In conclusion, our data clearly demonstrate that Eb1 and Msps require each other to
achieve prominent MT plus-end localisation in axons: Msps likely maintains Eb1 at MT plus-
ends through promoting GTP-cap formation, as is supported by our findings with GTP-tubu-
lin staining. *Vice versa*, Eb1-mediated recruitment of Msps might involve roles of Eb1 in the
structural maturation of MT plus-ends (see Discussion; [14,62]).

## Tau promotes Eb1 pools at MT plus-ends by outcompeting it from lattice binding

Similar to Msps deficiency, we found that also loss of Tau leads to a reduction of Eb1 comet
amounts in both *Drosophila* (Fig 1C and 1J and S2A and S2B Fig) and *Xenopus* neurons (S5C
and S5D Fig). In wild-type fly neurons, Tau localises along MT lattices and appears not to
extend into the Eb1 comet at the MT plus-end (Fig 5A' and 5A"). This distribution is consis-
tent with reports that Tau and Eb1 do not overlap at MT plus-ends due to their respective
higher affinities for GDP- and GTP-tubulin [69,70], suggesting that Tau promotes Eb1 plus-
end localisation through indirect mechanisms not involving their physical interaction.

We noticed that the reduction of Eb1 comet sizes in Tau-deficient neurons is accompanied
by a 20% increase in the intensity of Eb1 along MT lattices (Fig 5C and 5E for 6 day pre-cul-
ture, S8A Fig for embryonic cultures). This effect is specific for *tau* and not observed in
$msps^{A/A}$ mutant neurons (S8A and S8B Fig). Tau has previously been shown to protect MTs
against Katanin-induced damage [71]. Increased MT repair upon loss of Tau could therefore
promote the incorporation of GTP-tubulin along the lattice which would, in turn, recruit Eb1
[72]. However, MT lattices in Tau-deficient neurons did not show obvious increases in GTP-
tubulin (S7B and S7D Fig), arguing against the repair hypothesis. In the same specimens,
GTP-tubulin amounts at MT plus-ends were clearly reduced, consistent with the observed Eb1
comet depletion in Tau-deficient neurons (Fig 1J *vs*. S7A–S7C Fig).

We observed that over-expression of Tau in wild-type neurons decreased Eb1 levels along
MT lattices even further (S7E Fig), and similar observations were reported *in vitro* with mam-
malian versions of the two proteins [73]. We hypothesised therefore that Tau may competi-
tively prevent lattice-binding of Eb1, as similarly observed for other MAPs [74]: if Tau is
absent, MT lattices would therefore turn into a sink for Eb1 and sequester Eb1 pools away from
MT plus-ends. Such a mechanism could be particularly relevant in narrow axons which display
a high relative density of MTs [1]. In support of our hypothesis, we found that $tau^{KO/KO}$ mutant
neurons overexpressing Eb1::GFP showed even greater Eb1 intensity along MT lattices, but
also replenished Eb1 amounts at MT plus-ends (Fig 5D–5F). This treatment was sufficient to
suppress Tau-deficient phenotypes, as reflected in the recovery of Eb1 comet dynamics
(improved velocity and lifetime) and strongly reduced MT curling ('Eb1-GFP' in Fig 5G–5I).

These effects were specific for Tau and Eb1: Firstly, no improvement of the $tau^{KO/KO}$
mutant phenotypes was observed when expressing Shot-Ctail::GFP (as used in Fig 1K and 1L)

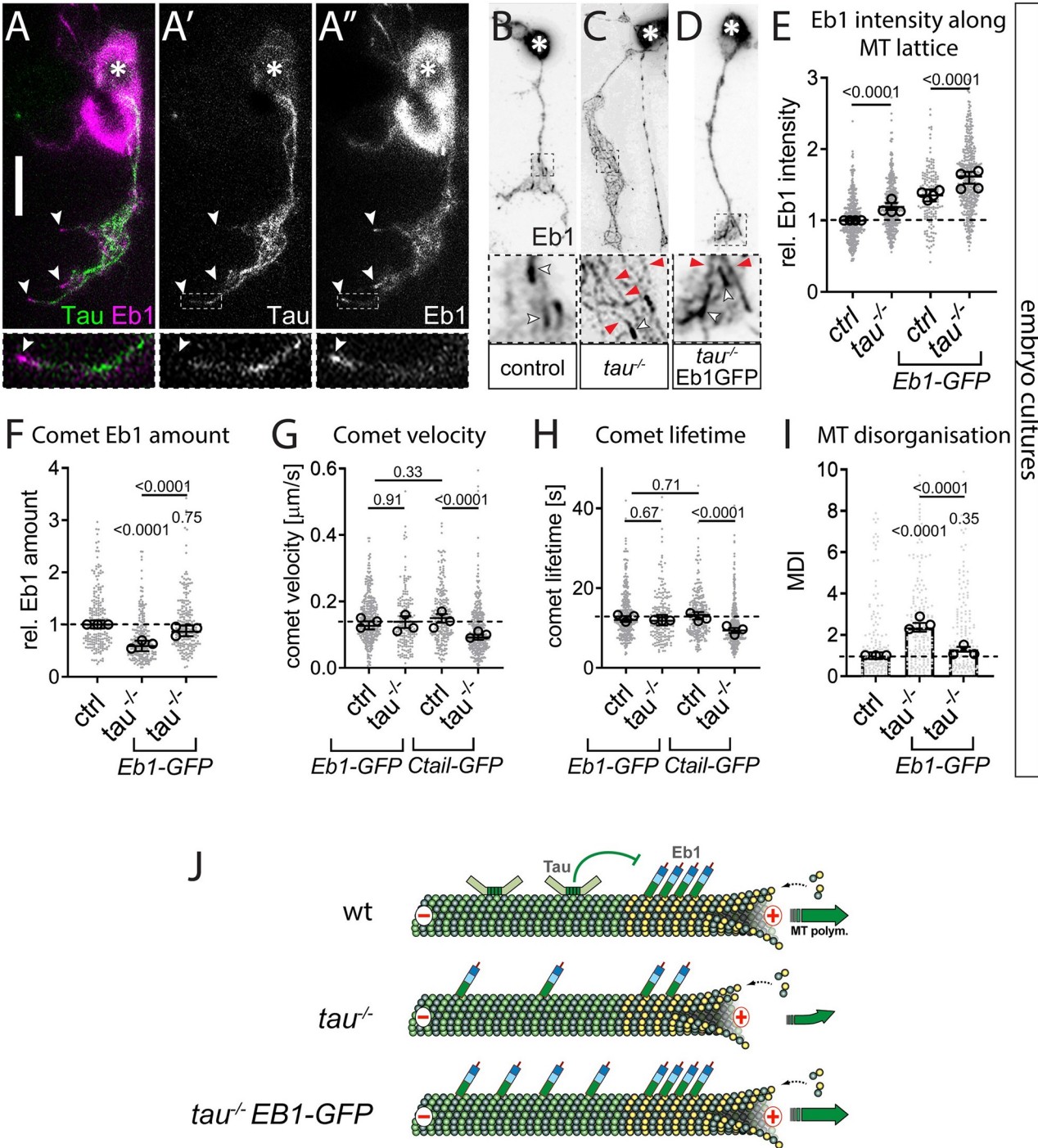

**Fig 5. Tau promotes Eb1 pools at MT plus-ends by outcompeting Eb1's association with the MT lattice. A-A")** Example of an embryonic neuron at 6 HIV imaged live with curling MTs to illustrate Tau binding (green) along the MT lattice, separated from Eb1 comets (magenta); asterisks indicate the soma, the scale bar represents 10 μm, dashed boxes indicate the positions of the 4-fold magnified close-ups shown at the bottom, white arrowheads point at Eb1 comets (same in B-D). **B-D)** Primary neurons at 6 HIV stained for Eb1 which are either wild-type (B) or mutant for $tau^{KO/Df}$ (C) or expressing Eb1::GFP driven by *elav-Gal4* in $tau^{KO/Df}$ mutant background (D); white arrowheads indicate Eb1 comets, red arrowheads Eb1 lattice localisation. **E-I)** Different parameters (as indicated) of control (ctrl) or $tau^{KO/Df}$ ($tau^{-/-}$) mutant neurons at 6 HIV without/with *elav-Gal4*-driven expression of Eb1::GFP or Shot-Ctail::GFP (as indicated); data were normalised to parallel controls (dashed horizontal lines) and are shown as scatter dot plots with mean ± SEM (I) or median ± 95% confidence interval (E-H) from at least two independent repeats with 3 experimental replicates; large open circles in graphs indicate median/mean of independent biological repeats. P-values obtained with Kruskal-Wallis ANOVA and Dunn's posthoc test as shown above data points/bars. **J)** Model view of the results shown here; note that yellow circles represent GTP-tubulin which provides higher affinity for Eb1 binding; for further explanations see main text and Discussion. For raw data see S5 Data.

to track MT plus-ends without altering Eb1 levels ('Ctail-GFP' in Fig 5G and 5H). Secondly, Eb1::GFP could not restore MT plus-end dynamics in $msps^{A/A}$ mutant neurons (S8D and S8E Fig), a finding that provides important insights also into the hierarchical relationships between Eb1, Msps and Tau (see Discussion).

In conclusion, we propose that Tau contributes to MT polymerisation dynamics and MT organisation indirectly, through preventing that Eb1 is sequestered away from MT plus-ends. The fact that Eb1::GFP can suppress *tau* mutant phenotypes including their MT curling, further highlights Eb1's crucial role in MT bundle promotion (compare Fig 2C and S4 Fig) and may suggest it even as a potential therapeutic target.

## An Eb1- and spectraplakin-dependent guidance mechanism explains roles of the functional trio in MT bundle organisation

Since Eb1 amounts at MT plus-ends inversely correlate with MT curling (Fig 2C and S4 Fig), bundle deterioration in either Eb1-, Msps- or Tau-deficient neurons might be a consequence of their negative impacts on Eb1 localisation. Eb1 at polymerising MT plus-ends has been proposed to recruit the C-terminus of Shot which simultaneously binds cortical F-actin via its N-terminus (Fig 6A). Through this cross-linking activity, Shot has been proposed to guide the extension of MTs along the axonal surface into parallel bundles (Fig 6F); accordingly, depletion of either Shot or Eb1 causes MT curling, and both show strong genetic interaction in this context ([21,22,75]; Fig 6F and S1D Fig).

In support of this guidance hypothesis, we found that severe MT curling observed in $Eb1^{+/-}$ $msps^{+/-}$ $tau^{+/-}$ triple-heterozygous mutant neurons (which have strongly reduced Eb1 comet amounts; Fig 2A") was significantly improved from 7.3-fold (with GFP-expression) to 1.4-fold, when over-expressing full length Shot-FL::GFP (Fig 6B and 6C; both compared to wild-type controls). This improvement could also be achieved through Eb1-independent roles of Shot in MT stabilisation mediated by the C-terminal Gas2-related domain (Fig 6A; [22]). We therefore repeated the experiment with two different Shot variants that maintain MT-stabilising activity whilst specifically abolishing actin-Eb1 cross-linkage (Fig 6A and 6F)): (1) Shot$^{\Delta ABD}$::GFP lacks the N-terminal calponin homology domain required for interaction with the actin cortex; (2) Shot$^{3MtLS*}$::GFP carries mutations in the C-terminal SxIP motifs required for Eb1 binding. Both these Shot variants failed to suppress MT curling in $Eb1^{04524/+}$ $msps^{1/+}$ $tau^{KO/+}$ triple-heterozygous mutant neurons (Fig 6B and 6C), hence lending further support to the guidance hypothesis based on Eb1-Shot interaction. That both proteins work in the same pathway is further substantiated by the finding that MT curling observed in $Eb1^{04524/04524}$ mutant neurons is not enhanced in $Eb1^{04524/04524}$ $shot^{3/3}$ double-mutant neurons, neither in normal embryonic nor in embryonic pre-cultured neurons (Fig 6D and 6E).

We conclude that co-operative promotion of MT polymerisation through Eb1, Msps and Tau appears to perform its additional function in axon bundle organisation through Shot-mediated MT guidance downstream of Eb1. In this way, the three proteins uphold the numbers as well as the bundled organisation of MTs as two key features underpinning the formation and long-term maintenance of axons.

## Discussion

### New understanding of the role and regulation of MT polymerisation and guidance in axons

Understanding the machinery of MT polymerisation is of utmost importance in axons where MTs form loose bundles that run along the neurite throughout its entire length; these bundles

 

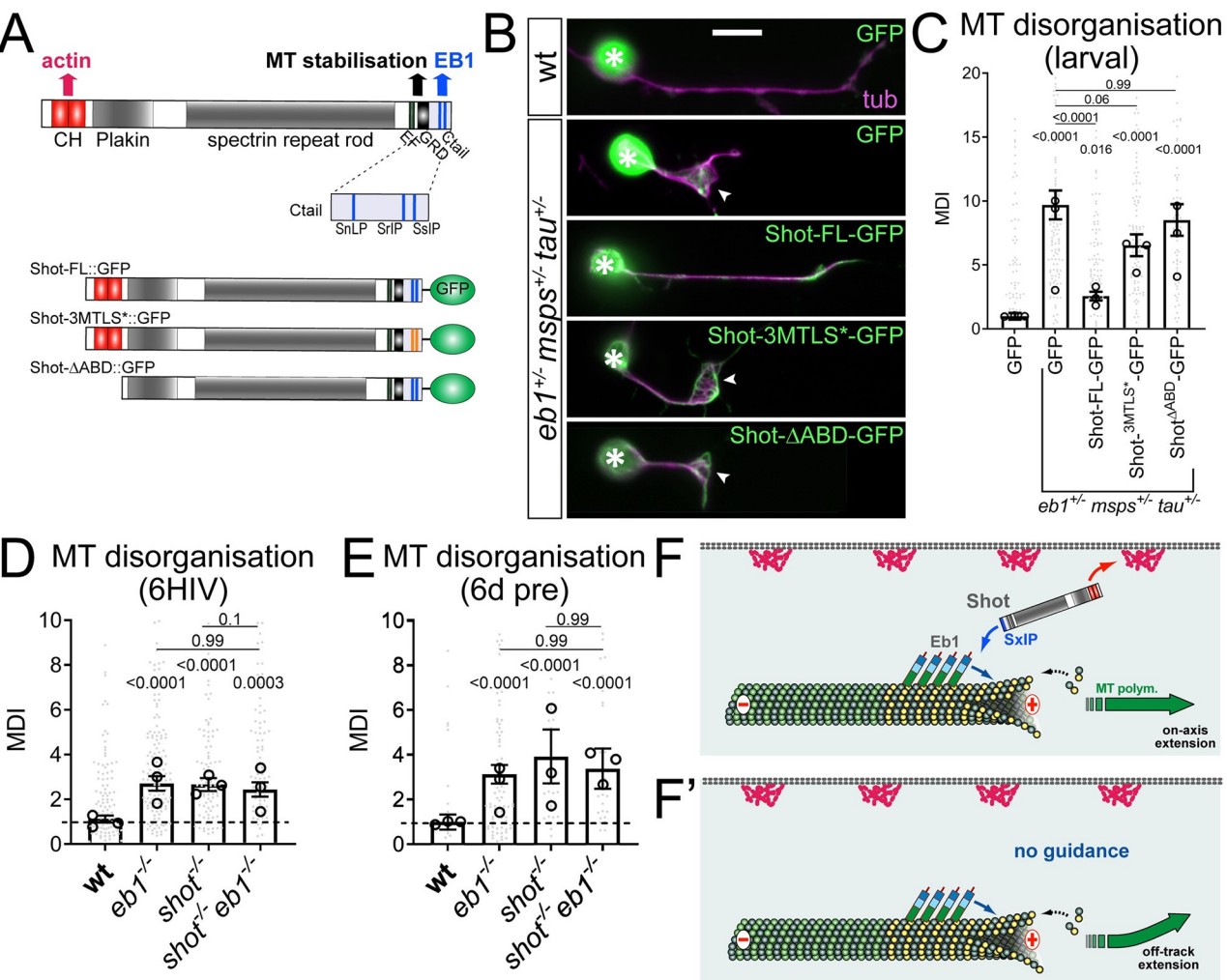

**Fig 6. Shot-mediated guidance mechanistically links Eb1 at MT plus-ends to bundle organisation. A**) Schematic representation of Shot constructs (CH, actin-binding calponin-homology domains; EF, EF-hand motifs; GRD, MT-binding Gas2-related domain; Ctail, unstructured MT-binding domain containing Eb1-binding SxIP motifs in blue); in Shot-3MtLS*::GFP the SxIP motifs are mutated (orange lines). **B**) Fixed primary neurons at 18HIV obtained from late larval CNSs stained for GFP (green) and tubulin (magenta), which are either wild-type (top) or *Eb1^{04524/+} msps^{A/+} tau^{KO/+}* triple-heterozygous (indicated on right) and express GFP or either of the constructs shown in D; scale bar 10μm. **C**) Quantification of MT curling of neurons as shown in B. **D,E**) MT curling in *shot^{3/3} Eb1^{04524/04524}* double-mutant neurons is not enhanced over single mutant conditions assessed in fixed embryonic primary neurons at 6HIV (D) or 12HIV following 6 day pre-culture (E). In all graphs data were normalised to parallel controls (dashed horizontal lines) and are shown as mean ± SEM from at least two independent repeats with 3 experimental replicates each; large open circles in graphs indicate median/mean of independent biological repeats. P-values obtained with Kruskall-Wallis ANOVA tests are shown above bars. **F,F'**) Model derived from previous work [22], proposing that the spectraplakin Shot cross-links Eb1 at MT plus-ends with cortical F-actin, thus guiding MT extension in parallel to the axonal surface; yellow dots represent GTP-tubulins providing high affinity sites for Eb1-binding. For raw data see S6 Data.

are essential for axonal morphogenesis and life-sustaining cargo transport, and must be maintained in functional state for up to a century in humans [1,2,6]. To achieve this, MT polymerisation is required to generate MTs *de novo*, repair or replace them. The underpinning machinery is expected to be complex [7], but deciphering the involved mechanisms will pay off by delivering new strategies for tackling developmental and degenerative axon pathologies [2].

Here we made important advances to this end. Having screened through 13 candidates (this work and [20]), we found the three factors Eb1, Msps and Tau to stand out by expressing

the same combination of phenotypes, and by displaying functional interaction in both *Drosophila* and *Xenopus* neurons. We found that their functions are not only important to maintain MT polymerisation, but also to align MTs into parallel arrangements, thus contributing in two ways to MT bundle formation and maintenance, both in culture and *in vivo*. The observed impact on MT organisation is also consistent with roles of XMAP215 during MT guidance in growth cones of frog neurons [56].

Our data reveal that various mechanisms observed *in vitro* or in non-neuronal cells, apply in the biological context of axons, which was unpredictable for two reasons: Firstly, of the three proteins only the human tau homologue has OMIM-listed links to human axonopathies, and these do not necessarily relate to MT polymerisation [17]. Absence of such disease links might well be due to the fact that these proteins are functionally too important, causing embryonic lethality when dysfunctional [2]. Secondly, axons and non-neuronal cells can display significant mechanistic deviations as shown for CLIP170/190 [20] and for the MT localisation of Msps which is facilitated by Sentin or dTACC in non-neuronal cells, dendrites and *in vitro* [15,63–65] but seemingly not in axons (S6 Fig). However, other mechanisms we observed in axons matched previous reports: (1) the complementary binding preferences of Eb1 and Tau for GTP-/GDP-tubulin [69,76]; (2) the mutual enhancement of Eb1 and XMAP215/Msps [14,15,77]; (3) the correlation of GTP cap size with comet velocity [58]. Furthermore, we observed that depletion of α1-tubulin in neurons mutant for *αtub84B* or *stathmin* (a promoter of tubulin availability; [25,57]) affects comet numbers but not Eb1 amounts (S1 Fig); this is consistent with observations that MT nucleation *in vitro* is far more sensitive to tubulin levels than polymerisation [78].

## Eb1 and XMAP215/Msps are core factors promoting MT polymerisation and guidance

Apart from demonstrating the relevance of various molecular mechanisms in the context of axonal MT regulation, our work provides key insights as to how they integrate into one consistent mechanistic model of biological function (summarised in Fig 7).

The TOG-domain protein XMAP215/Msps is relevant for neuronal morphogenesis in fly and *Xenopus* [47,65], likely through its expected function as a MT polymerase [11,14,15,67,79,80]. In contrast, *Drosophila* and vertebrate Eb proteins are only moderate promoters of MT polymerisation *in vitro* ([14,15,73] and references within), but rather act as scaffolds [81]. Conserved binding partners of Eb proteins are the spectraplakins which can guide extending MT plus-ends along actin networks in axons and non-neuronal cells [22,75,82].

We propose therefore (see Fig 7A) that Eb1 is the key mediator of MT guidance into bundles (as supported by data throughout this work; Figs 2C and 6 and S4 Fig), and Msps the key promoter of MT polymerisation (Fig 4E and S8 Fig). To execute these functions, both proteins depend on each other: MT plus-end localisation of Msps is reduced upon loss of Eb1 (Figs 4 and 7B) and *vice versa* (Figs 1B and 1J and 7C and S1B Fig). This mutual dependency is unlikely to involve their physical interaction, since MT plus-end localisation of Eb1 is known to occur tens of nanometres behind XMAP215 [14,62], as seems to be the case also for axonal MTs (Fig 4A). Furthermore, our data do not support an obvious role of the Sentin or dTACC adaptors in mediating Eb1-XMAP215 interactions (Fig 4D–4D" and S6 Fig).

Potential indirect mechanisms explaining this co-dependency are provided by the promotion of MT polymerisation through XMAP215/Msps which maintains a prominent GTP-cap (S8C Fig) that, in turn, mediates Eb1 binding (see also [14,62]; Fig 7C). Restricted GTP-cap formation as a limiting factor for Eb1 binding would also explain why Eb1 over-expression fails to improve Msps-deficient phenotypes (S8D–S8F Fig). *Vice versa*, Eb proteins promote

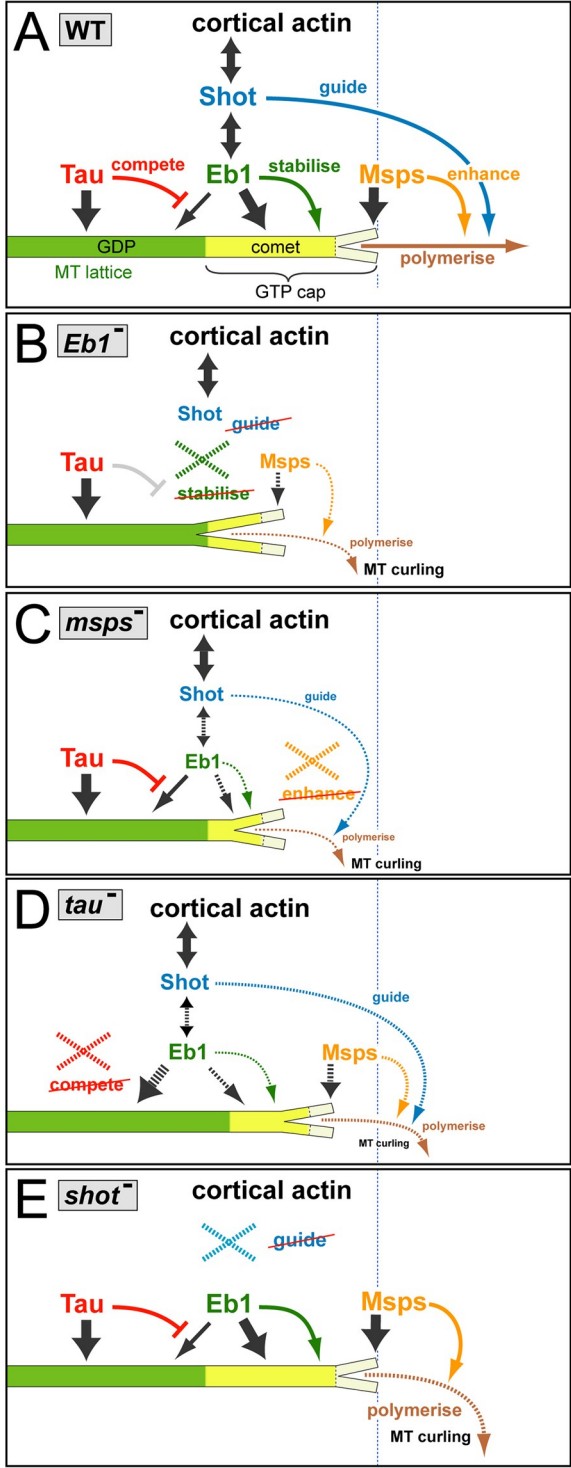

**Fig 7. A mechanistic model consistent with all reported data. A**) In wild-type (WT) neurons, the three factors bind (dark grey arrows) to MTs at different locations: Msps binds to the very tip of MT plus-ends, Eb1 forms plus-end comets (bright yellow) by binding with higher affinity to GTP-tubulin (GTP-cap, curly bracket) but lagging behind the front, and Tau localises along the MT lattice primarily composed of GDP-tubulin (green). Tau outcompetes (red T-bar) low-affinity binding of Eb1 along the lattice, thus maintaining more Eb1 at MT plus-ends. Msps enhances (orange arrow) MT polymerisation (brown), thus sustaining a prominent GTP-cap that Eb1 can bind to. Eb1 stabilises MT plus-ends (green arrow) by promoting sheet formation of protofilaments at the plus-end (short Y-shaped tip), thus improving conditions for Msps binding. Eb1 also recruits the C-terminus of Shot which binds cortical actin via its N-

terminus, thus establishing cross-linkage that can guide (blue arrow) extending MT plus-ends into parallel bundles. **B-E**) Illustrations explaining the changes triggered by the loss of the different factors (stippled X); any affected processes are shown as stippled lines with reduced thickness and reduced font sizes of accompanying texts. **B**) Upon loss of Eb1, the plus-end sheet structure is weakened (larger Y shape), thus reducing Msps binding and, in turn, reducing polymerisation; Shot detaches, thus abolishing guidance. **C**) Loss of Msps abolishes enhanced polymerisation so that the GTP-cap shrinks and less Eb1 binds, thus also weakening Shot binding and MT guidance. **D**) Upon loss of Tau, more Eb1 can bind to the MT lattice, thus reducing the amounts available for plus-end association; this causes a modest Eb1 depletion phenotypes with consequences similar to B, but less pronounced. **E**) Upon loss of Shot, the localisations and functions of the other three proteins are unaffected, but MT guidance is abolished.

lateral protofilament contacts which could assist in sheet formation at the very plus tip (Fig 7B), thus facilitating the binding of XMAP215/Msps [14,62].

## Tau contributes through an indirect mechanism of competitive binding to MT lattices

Tau and Map1b/Futsch are known to bind along MT lattices, to promote MT polymerisation *in vitro*, and to enhance axon growth in mouse and fly neurons through mechanisms that remain unclear [12,29,48,73,83–93].

In our cellular model, loss of the Map1b homologue Futsch has no obvious effects, whereas Tau shares all assessed loss-of-function mutant phenotypes with Msps and Eb1, although with weaker expression. Of these phenotypes, reduced MT plus-end localisation of Eb proteins upon loss of Tau function (Fig 1C and 1J) was likewise reported for frog neurons (S5C and S5D Fig), N1E-115 mouse neuroblastoma cells and primary mouse cortical neurons [94].

One proposed mechanism involves direct interaction where Tau recruits Eb proteins at MT plus ends [94], consistent with other reports that Tau can bind Eb1 [70,95]. However, further reports argue against overlap of Tau and Eb1 and rather show that Eb1 and Tau have complementary preferences for GTP- and GDP-tubulin, respectively [69,70,76]; this is also consistent with our data (Fig 5). Furthermore, a recruitment model is put in question by our finding that Eb1 lattice localisation increases rather than decreases upon loss of Tau (Fig 5E), as similarly observed for mammalian tau and Eb1 *in vitro* [73].

Therefore, we propose a different mechanism based on competitive binding where Tau's preferred binding to GDP-tubulin along the lattice prevents Eb1 localisation (Fig 5), comparable to Tau's role in preventing other proteins including MAP6 and MAP7 from binding in certain regions of the MT lattice [74,96–98]. Given the high density of MTs especially in small-diameter axons [1], lattice binding could generate a sink large enough to reduce Eb1 levels at MT plus-ends, and our experiments with Eb1::GFP overexpression strongly support this notion (Fig 5). In this way, loss of Tau generates a condition comparable to a mild Eb1 loss-of-function mutant phenotype, thus explaining why Tau shares its repertoire of loss-of-function phenotypes with Msps and Eb1, but with more moderate presentation. This competition mechanism might apply in axons with high MT density, for example explain the reduction in axonal MT numbers upon Tau deficiency in *C. elegans* [99]; it might be less relevant in larger diameter axons of vertebrates where MT densities are low [1].

## Main conclusions and future perspectives

Here we have used a standardised *Drosophila* neuron system amenable to combinatorial genetics to gain understanding of MT regulation at the cellular level in axons. We propose a consistent mechanistic model that can integrate all our data, mechanisms reported in the literature, and our previous mechanistic model explaining Eb1/Shot-mediated MT guidance [22,37].

This understanding offers new opportunities to investigate the mechanisms behind other important observations.

For example, the presence of an axonal sleeve of cortical actin/spectrin networks was shown to be important to maintain MT polymerisation, likely relevant in certain axonopathies [41]; the underlying mechanisms are now far easier to dissect. As another example, we found that loss of either Eb1, XMAP215/Msps or Tau all caused a reduction in comet numbers, consistent with reports of nucleation-promoting roles of XMAP215 in non-neuronal contexts [100–104] or reactivating neuronal stem cells [105]. This might offer opportunities to investigate how axonal MT numbers can be determined through the regulation of local acentrosomal nucleation in reproducible, neuron/axon-specific ways, thus addressing a fundamental aspect of axon morphology [1].

By gradually assembling molecular mechanisms into regulatory networks that can explain axonal MT regulation at the cellular level, i.e. the level at which diseases become manifest, our studies come closer to explaining axonal pathologies which can then form the basis for the development of remedial strategies [6].

## Supporting information

**S1 Fig. A candidate screen of axonal loss-of-function phenotypes in primary neurons.** Graphs show extended data sets for four of the parameters displayed in Fig 1 (indicated above each graph). Data points/bars representing mutant conditions for different genes are consistently colour-coded in all graphs, and conditions used are indicated below (6HIV, cultured from embryos for 6hrs; 6d/7d pre, cultured from embryos for 12hrs following 6 or 7 days pre-culture; L3, cultured from late larval CNS for 18hrs). Allele names are given as superscript: absence of slash indicates homozygous, presence of slash hetero-allelic conditions. Data were normalised to parallel controls (dashed horizontal line) and are shown as median ± 95% confidence interval (B,C) or mean ± SEM (A,D); data points from at least two experimental repeats consisting of 3 replicates each are shown, large open circles in graphs indicate median/mean of independent biological repeats. P-values obtained with Kruskall-Wallis ANOVA test above data points/bars. For raw data see S7 Data.
(TIF)

**S2 Fig. Correlation of different Eb1 comet properties. A,B**) Eb1 amount at comets is calculated as the product of comet length (A) and the fluorescent mean intensity of Eb1 comets (B), which are both affected to similar degrees by homozygous condition of $msps^A$, $tau^{KO}$ and $Eb1^{04524}$ in embryo-derived neurons cultured for 12hrs following 6 day pre-culture (6d pre); data were normalised to controls (dashed horizontal line) and are shown as scatter dot plot with median ± 95% confidence interval of at least three experimental repeats, large open circles in graphs indicate median/mean of independent biological repeats. P-values listed above each plot were obtained with Kruskall-Wallis ANOVA tests. **C**) The table lists data for Eb1 amounts (fixed neurons; compare Fig 1A–D and 1J) or for comet velocity/lifetime (live imaging; compare Fig 1K and 1L), all obtained from pre-cultured embryonic primary neurons carrying the same combinations of mutant alleles in homozygosis (indicated on the left; used alleles: $tub84B^{def}$, $msps^A$, $tau^{KO/Df}$, $eb1^{04524}$, $stai^{KO}$). **D,E**) Plotting comet velocity or lifetime against Eb1 amounts from different genetic conditions shows good correlation (r and p-value determined via non-parametric Spearman correlation analysis). For raw data see S8 Data.
(TIF)

**S3 Fig. Comet numbers and dynamics in (trans-) heterozygous conditions. A**) Eb1 comet numbers in fixed primary neurons cultured for 12hrs following 5 day pre-culture. **B**) Comet

velocity and lifetime obtained from live analyses of primary neurons cultured for 18hrs from CNSs of late larvae carrying single-, double- or triple-heterozygous conditions (complementing data in Fig 2A). In all graphs, data were normalised to parallel controls (dashed horizontal lines) and are shown as scatter dot plots with median ± 95% confidence interval (B) or bar chart with mean ± SEM (A) of at least two experimental repeats; large open circles in graphs indicate median/mean of independent biological repeats. P-values obtained with Kruskal-Wallis ANOVA test are given above data points/bars; used alleles: $msps^A$, $tau^{KO}$, $Eb1^{04524}$. For raw data see S9 Data.
(TIF)

**S4 Fig. Increased Eb1 amounts correlate with decreased MT curling.** The data and graph show further details behind the correlations displayed in Fig 2C. **A**) The table provides descriptions, data and references for the graph shown in B: coloured numbers (1st column) correspond to numbers of data points in B; the different allelic combinations and culture conditions used (2nd column) comprise embryonic neurons cultures for 6 hrs (6HIV), embryonic neurons cultured for 12 hrs following 6 day preculture (6d pre), neurons cultured from larval CNSs for 18hrs (L3); respective data for Eb1 amounts (3rd column) and MT curling (4th column) were obtained from different sets of experiments throughout this work (5th column lists the figures from where these data originate). **B**) Correlation plot of the data shown in A, with numbers and colours of data points corresponding to the 1st column; r and p-value determined via non-parametric Spearman correlation analysis. For raw data see S10 Data.
(TIF)

**S5 Fig. Support data for *Xenopus* experiments. A-B')** A RT-PCR DNA gel (A) and Western blot (B) and their quantifications (A', B') show the degrees of EB3/Tau knock-downs upon application of different morpholino concentrations (indicated on top in blots and at the bottom in graphs); ODC1 and ß-actin are used as loading controls; data are normalised to no-morpholino controls from two experimental repeats (dashed lines). 50% knock-down of XMAP215 was achieved by injecting 6 ng of the validated XMAP215 MO as described previously [47,56]. **C-D**) Different properties of MACF43::GFP comets (as indicated upon graphs) obtained from *Xenopus* primary neurons, either upon 50%; data were normalised to parallel controls (dashed horizontal lines) and are shown as median ± 95% confidence interval; merged sample numbers from at least two experimental repeats are shown at the bottom, P-values obtained with Kruskall-Wallis ANOVA and Dunn's posthoc tests above data points. For raw data see S11 Data.
(TIF)

**S6 Fig. Loss of TACC or Sentin does not cause obvious axonal phenotypes.** Axon length (**A**) and MT curling (**B**) embryonic 6d pre-cultured neurons which were either wild-type (wt) or homozygous mutant for *sentin* or *dTACC* (as indicated); data were normalised to parallel controls (dashed horizontal lines) and are shown as scatter dot plots with median ± 95% confidence interval (A) or mean ± SEM (B) from at least two experimental repeats; large open circles in graphs indicate median/mean of independent biological repeats. P-values obtained with Kruskal-Wallis ANOVA tests are shown above data points/bars. For raw data see S12 Data.
(TIF)

**S7 Fig. In Tau-deficient axons, GTP-tubulin is reduced at MT plus-ends but unchanged at lattices. A-B'')** Fixed primary neurons stained for Eb1 (magenta) and GTP-tubulin (green; the scale bar represents 10μm); asterisks indicate somata, dashed boxes the positions of the 4-fold magnified close-ups shown at the bottom, arrowheads point at Eb1::GFP comets and GTP

caps. **C,D**) Graphs showing staining intensity of GTP-tubulin at MT plus-ends (C) and along the MT lattice (D) of embryonic neuron at 6 HIV. **E)** Graphs showing staining intensity of Eb1 along the MT lattice of neurons without/with *elav-Gal4*-driven expression of *Drosophila* Tau (*dtau*). Overexpression of *dtau* leads to a reduction of Eb1 at the MT shaft; data were normalised to parallel controls (dashed horizontal lines) and are shown as scatter dot plots with median ± 95% confidence interval from at least two independent repeats with 3 experimental replicates; large open circles in graphs indicate median/mean of independent biological repeats. P-values obtained with Kruskall-Wallis ANOVA test above data points/bars. For raw data see S13 Data.
(TIF)

**S8 Fig. Details of Eb1-Msps cross-regulation. A,B**) Eb1 shaft localisation is unaffected by loss of Msps in primary neurons at 6HIV (A) or at ~12HIV following 6 day pre-culture (B). **C)** Staining intensity of GTP-tubulin at MT plus-ends is reduced in $msps^{A/A}$ and $Eb1^{04524/04524}$ mutant embryonic neurons at ~12 HIV following 6 day pre-culture. **D-F**) Expressing Eb1:: GFP via *elav-Gal4* does not improve comet velocities (D), comet lifetime (E) and MT curling (F) in primary neurons of $msps^{A/146}$ mutants (cultured12 HIV following 6 day pre-culture); data were normalised to parallel controls (dashed horizontal lines) and are shown as scatter dot plots with mean ± SEM (F) or median ± 95% confidence interval (A-E) from at least two experimental repeats; large open circles in graphs indicate median/mean of independent biological repeats. P-values obtained with Kruskal-Wallis ANOVA tests and Dunn's posthoc analyses are shown above data points/bars. For raw data see S14 Data.
(TIF)

**S1 Data. Raw data files for Fig 1.**
(XLSX)

**S2 Data. Raw data files for Fig 2.**
(XLSX)

**S3 Data. Raw data files for Fig 3.**
(XLSX)

**S4 Data. Raw data files for Fig 4.**
(XLSX)

**S5 Data. Raw data files for Fig 5.**
(XLSX)

**S6 Data. Raw data files for Fig 6.**
(XLSX)

**S7 Data. Raw data files for S1 Fig.**
(XLSX)

**S8 Data. Raw data files for S2 Fig.**
(XLSX)

**S9 Data. Raw data files for S3 Fig.**
(XLSX)

**S10 Data. Raw data files for S4 Fig.**
(XLSX)

**S11 Data. Raw data files for S5 Fig.**
(XLSX)

**S12 Data. Raw data files for S6 Fig.**
(XLSX)

**S13 Data. Raw data files for S7 Fig.**
(XLSX)

**S14 Data. Raw data files for S8 Fig.**
(XLSX)

**S1 Movie. Msps::GFP and Eb1::RFP jointly track MT plus-ends.** Live movie of a wild-type neuron co-expressing Msps::GFP and Eb1::RFP; for stills see Fig 4A–4A". As indicated, single channels are shown on the left and middle, and the combined movie on the right. The movie was acquired at 0.5 frames per second and plays at 0.5 s per frame. The scale bar indicates 10 μm.
(GIF)

**S2 Movie. Msps plus-end localisation in wild-type neurons.** Live movie of a wild-type neuron expressing Msps::GFP; for a still see Fig 4C. The movie was acquired at 1 frame per second and plays at 0.2 s per frame. The scale bar indicates 10 μm.
(GIF)

**S3 Movie. Msps plus-end localisation is impaired in the absence of Eb1.** Live movie of an $Eb1^{04524/04524}$ mutant neuron expressing Msps::GFP; for a still see Fig 4C'. The movie was acquired at 1 frames per second and plays at 0.2 s per frame. The scale bar indicates 10 μm.
(GIF)

## Acknowledgments

We thank Hiro Ohkura for kindly providing DmEb1 antibody and mutant alleles of *msps*.

## Author Contributions

**Conceptualization:** Ines Hahn, Natalia Sanchez-Soriano, Andreas Prokop.

**Data curation:** Ines Hahn, Andre Voelzmann, Paula G. Slater, Natalia Sanchez-Soriano.

**Formal analysis:** Ines Hahn, Andre Voelzmann, Jill Parkin, Judith B. Fülle, Paula G. Slater, Natalia Sanchez-Soriano.

**Funding acquisition:** Ines Hahn, Andre Voelzmann, Paula G. Slater, Laura Anne Lowery, Natalia Sanchez-Soriano, Andreas Prokop.

**Investigation:** Ines Hahn, Judith B. Fülle, Paula G. Slater, Natalia Sanchez-Soriano.

**Methodology:** Ines Hahn, Jill Parkin, Paula G. Slater, Natalia Sanchez-Soriano.

**Project administration:** Ines Hahn, Andreas Prokop.

**Supervision:** Ines Hahn, Laura Anne Lowery, Andreas Prokop.

**Validation:** Ines Hahn, Paula G. Slater, Natalia Sanchez-Soriano.

**Visualization:** Ines Hahn, Andre Voelzmann, Andreas Prokop.

**Writing – original draft:** Ines Hahn, Natalia Sanchez-Soriano, Andreas Prokop.

**Writing – review & editing:** Ines Hahn, Andre Voelzmann, Paula G. Slater, Laura Anne Lowery, Natalia Sanchez-Soriano, Andreas Prokop.

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
