## [Decision Letter · Decision Letter 0]

12 May 2021

Dear Dr Hahn,

Thank you very much for submitting your Research Article entitled 'Tau, XMAP215/Msps and Eb1 co-operate interdependently to regulate microtubule polymerisation and bundle formation in axons' to PLOS Genetics.The manuscript was fully evaluated at the editorial level and by three independent peer reviewers. The reviewers appreciated the attention to an important topic but identified some concerns that we ask you address in a revised manuscript

We therefore ask you to modify the manuscript according to the review recommendations. Your revisions should address the specific points made by each reviewer.

[LINK]

Yours sincerely,

Fengwei Yu

Associate Editor

PLOS Genetics

Gregory P. Copenhaver

Editor-in-Chief

PLOS Genetics

As you will see, all referees acknowledge that the findings are interesting and important. Referees only raised some minor points. I think all comments are helpful and should be addressed in the revised manuscript.

Reviewer's Responses to Questions

**Comments to the Authors:**

Reviewer #1: This is a fascinating treatment of how three very different proteins cooperate functionally in the neuron. The treatment is complicated to sort through, but an important and original way to understand functionality that is much deeper than simply knocking out one protein and then interpreting the phenotype without taking into account other players that interact together. I believe the data could be interpreted in other ways than the authors do, for example with the depletion of any of the three players impacting microtubule stability in a manner that affects the other two players. However, I like their bold and original stance, and I think it behooves the community to consider this approach without waffling on it. I have no suggestions for changes.

Reviewer #2: Understanding the mechanisms of MT polymerization in axons in vivo is of great importance to decipher the developmental and degenerative axon pathologies. In this manuscript, Hahn et al. identified the role of the microtubule (MT) lattice binder Tau, microtubule regulator Msps, and end-binding protein EB1 during axonal microtubule polymerization in both Drosophila and Xenopus neurons. The authors further discovered the interdependency among the trio during axonal transport. They showed that EB1 and XMAP215/Msps work interdependently at MT plus-ends, whereas Tau acts through outcompeting EB1-binding along MT lattices and therefore maintains EB1 pools at MT plus-ends. They have carried out comprehensive analysis on microtubule organization in axonal MT growth, including axonal length, EB1 comet amount, EB1 comet intensity, comet lifetime, and MT curling. Convincing data with proper statistical analysis are provided throughout the manuscript. Finally, they proposed that the trio promotes the bundle conformation of axonal MTs through a guidance mechanism by the spectraplakin shot.

Overall, this study provided new significant advancement to the mechanism of MT polymerization and organization in axons. The manuscript has shown a large amount of data and is well written. This study will clearly attract general audience of PLOS Genetics. I am supportive of the manuscript acceptance once the following minor concerns have been addressed.

-The function of Msps in microtubule polymerization was analyzed extensively in dividing cells in which centrosomes serve as the microtubule organizing center. In mature neurons, centrosomes are immature and acentrosomal microtubules are organized in both axons and dendrites. It would worth a discussion to emphasize the novelty of this study in revealing the role of Msps in organizing acentrosomal microtubules for axonal transport. It would also be good to cite the following three papers on the role of Msps on acentrosomal microtubule regulation in various cell types.

Quan Tang et al., EMBO J 2020;

Yiming Zheng et al., Nature Cell Biol. 2020;

Qiannan Deng et al., BioRxiv 2020.

-“Rescue” needs to be used more accurately. When introducing a different gene to a mutant/RNAi condition, “suppression” should be used formally, as “rescue” often refers to the result by re-introducing the same gene that was mutated/depleted in the mutant/RNAi.

- The fact that variants in CKAP5/XMAP215 and EB1 have not been identified in human patients with neurological disorders might be due to the essential function of CKAP5 and/or EB1 during embryonic development. The author might want to consider this possibility when discussing the disease implications.

-Define HIV in the first place.

-The references on line 81 need to be formatted.

Reviewer #3: In their manuscript, Hahn et al. perform a candidate screen to identify the core regulatory machinery involved in microtubule polymerization dynamics in axons. In analyzing a number of parameters, they observe similar phenotypes for loss of EB1, msps (XMAP), and tau (though tau has the mildest effect of the three) in fly neurons. The authors then perform an impressive array of experiments to analyze microtubule dynamics and axon length with every possible genetic combination of these three genes in Drosophila larval and embryonic cultures, Drosophila brains, AND in Xenopus primary neurons. These sets of experiments show that not only are these three proteins functionally related, but that their functional interplay is evolutionarily conserved. EB1 and msps depend on one another for proper localization and function at the microtubule plus ends, while tau outcompetes EB1 from the GDP lattice to ensure specific localization of EB1 at the polymerizing plus ends. They also show that this three-protein unit promotes bundling of axonal microtubules through a guidance mechanism mediated by Shot (a spectraplakin). Overall, this is an impressive paper with an immense amount of data, but it is organized and explained very nicely. In addition the authors do an excellent job of discussing prior work and integrating their results within the context of the field to develop the most comprehensive model to date. There are some additional discussion points and a couple of experiments that could add to the paper, but the additional experiments are not necessary for publication of this manuscript in Plos genetics.

1) The results with EB1 facilitating msps recruitment without the necessity of other adaptors is very interesting and the overall the result is very nicely discussed and placed in the context of the current literature in the field. The authors clearly show that the CTD is dispensible, indicating that adaptors are not necessary to recruit msps to MT plus-ends in fly neurons. It would be nice for the authors to discuss these results a bit in the discussion (page 13 or 14), to substantiate their claim that there is not a direct interaction between msps and eb1 (or any other adaptor), but a through-lattice facilitation of the binding of one another.

2) An interesting experiment/control would be to analyze EB1 intensity on the microtubule lattice upon overexpression or knockdown of DCX-EMAP, especially since the binding sites are more similar. This may prove to be difficult depending on which fly lines are available and is not necessary for the publication of this paper, but it would be interesting if DCX were playing a similar or different role with regards to effects on EB1.

3) In the last paragraph of the tau discussion on page 14, tau has also been shown to prevent MAP7 from binding the same region of the lattice (Monroy et al., Nat Comm, 2018), similarly to how it prevents MAP6 and now EB1. Citing this paper would help support the more global role for tau in this function. In addition, the authors should cite the work on tau by Tan et al., NCB, 2019 showing that tau forms oligomeric assemblies on the microtubule that are capable of blocking other MAPs, motors, and severing enzymes. This shows a clear mechanism by which tau is able to dictate access of other proteins to the microtubule lattice, lending further support for the observations in this paper.

4) There are some minor typos throughout, such as Fig 2: MT disorganisaiton should be “disorganisation”.

**Have all data underlying the figures and results presented in the manuscript been provided?**

Reviewer #1: Yes

Reviewer #2: Yes

Reviewer #3: Yes

PLOS authors have the option to publish the peer review history of their article (what does this mean?). If published, this will include your full peer review and any attached files.

Reviewer #1: **Yes: **Peter W. Baas

Reviewer #2: No

Reviewer #3: No

---

## [Decision Letter · Decision Letter 1]

7 Jun 2021

Dear Dr Hahn,

We are pleased to inform you that your manuscript entitled "Tau, XMAP215/Msps and Eb1 co-operate interdependently to regulate microtubule polymerisation and bundle formation in axons" has been editorially accepted for publication in PLOS Genetics. Congratulations!

Yours sincerely,

Fengwei Yu

Associate Editor

PLOS Genetics

Gregory P. Copenhaver

Editor-in-Chief

PLOS Genetics

Comments from the reviewers (if applicable):

Reviewer's Responses to Questions

**Comments to the Authors:**

Reviewer #1: Good job on the revisions.

Reviewer #2: The authors have adequately addressed my concerns. I now recommend the publication of this manuscript in PLOS Genetics.

Reviewer #3: The authors have addressed all of my concerns. I support publication in Plos Genetics.

**Have all data underlying the figures and results presented in the manuscript been provided?**

Reviewer #1: Yes

Reviewer #2: Yes

Reviewer #3: None

PLOS authors have the option to publish the peer review history of their article (what does this mean?). If published, this will include your full peer review and any attached files.

Reviewer #1: **Yes: **Peter W. Baas

Reviewer #2: No

Reviewer #3: No

**Data Deposition**

http://datadryad.org/submit?journalID=pgenetics&manu=PGENETICS-D-21-00546R1

**Press Queries**

---

## [Editor Report · Acceptance letter]

30 Jun 2021

PGENETICS-D-21-00546R1 

Tau, XMAP215/Msps and Eb1 co-operate interdependently to regulate microtubule polymerisation and bundle formation in axons 

Dear Dr Hahn, 

We are pleased to inform you that your manuscript entitled "Tau, XMAP215/Msps and Eb1 co-operate interdependently to regulate microtubule polymerisation and bundle formation in axons" has been formally accepted for publication in PLOS Genetics! Your manuscript is now with our production department and you will be notified of the publication date in due course.

With kind regards,

Zsofi Zombor

PLOS Genetics

On behalf of:
